# Dynamic transcriptomes identify biogenic amines and insect-like hormonal regulation for mediating reproduction in *Schistosoma japonicum*

Jipeng Wang[1,2,*], Ying Yu[1,*], Haimo Shen[2], Tao Qing[1], Yuanting Zheng[1], Qing Li[1], Xiaojin Mo[2], Shuqi Wang[1], Nana Li[1], Riyi Chai[1], Bin Xu[2], Mu Liu[1], Paul J. Brindley[3], Donald P. McManus[4], Zheng Feng[2], Leming Shi[1] & Wei Hu[1,2]

Eggs produced by the mature female parasite are responsible for the pathogenesis and transmission of schistosomiasis. Female schistosomes rely on a unique male-induced strategy to accomplish reproductive development, a process that is incompletely understood. Here we map detailed transcriptomic profiles of male and female *Schistosoma japonicum* across eight time points throughout the sexual developmental process from pairing to maturation. The dynamic gene expression pattern data reveal clear sex-related characteristics, indicative of an unambiguous functional division between males and females during their interplay. Cluster analysis, *in situ* hybridization and RNAi assays indicate that males likely use biogenic amine neurotransmitters through the nervous system to control and maintain pairing with females. In addition, the analyses indicate that reproductive development of females involves an insect-like hormonal regulation. These data sets and analyses serve as a foundation for deeper study of sexual development in this pathogen and identification of novel anti-schistosomal interventions.

[1] State Key Laboratory of Genetic Engineering, Ministry of Education Key Laboratory of Contemporary Anthropology, Collaborative Innovation Center for Genetics and Development, School of Life Sciences, Fudan University, Shanghai 200438, China. [2] Key Laboratory of Parasite and Vector Biology of the Chinese Ministry of Health, WHO Collaborating Center for Malaria, Schistosomiasis and Filariasis, National Institute of Parasitic Diseases, Chinese Center for Disease Control and Prevention, Shanghai 200025, China. [3] Department of Microbiology, Immunology & Tropical Medicine, Research Center for the Neglected Diseases of Poverty, School of Medicine & Health Sciences, George Washington University, Washington DC 20037, USA. [4] Molecular Parasitology Laboratory, Immunology Department, QIMR Berghofer Medical Research Institute, Brisbane, Queensland 4006, Australia. * These authors contributed equally to this work. Correspondence and requests for materials should be addressed to W.H. (email: huw@fudan.edu.cn).

Schistosomiasis is a neglected tropical disease affecting up to 250 million people in 76 countries[1]. It is caused by infection with worms of the trematode genus *Schistosoma*. The clinical symptoms and spread of this disease are due mainly to the eggs produced by the female parasites. Notably, schistosomes possess a reproductive mode that involves continuous pairing of the female with the male in the gynecophoral canal of the male to ensure sexual development and to maintain the mature state of the female gonad[2,3]. During pairing, germ cells in the ovary and vitelline gland commence differentiation and produce oocytes or vitellocytes, respectively[2]. Although a chemical or tactile stimulus from the male is thought to be transferred to the female which triggers a cascade of changes during the pairing process, the stimulating factor(s) or mechanisms associated with male-induced female reproductive development have not been elucidated[2,4–17]. Understanding the male–female interactions and their effects on reproductive development will provide new insights for the development of novel anti-schistosome strategies.

Transcriptomic analysis can be used to determine gene expression profiles that occur during development or under different environmental conditions, thereby facilitating further understanding of the development process and its potential regulatory mechanisms. RNA-seq is an insightful, powerful transcriptomics tool capable of delivery of genome-scale transcription profiles unconstrained by genomic annotation[18–22]. Transcriptomic studies by microarray or RNA-seq have been performed with mature and immature females, and with different sexes in *Schistosoma japonicum* and *S. mansoni*. Compared with immature females, genes involved in egg productions and haemoglobin digestions are highly enriched in adult females[23–27], whereas the expression of genes associated with the structure of the tegument and movement are enriched in adult males compared with adult females[28–33]. Nevertheless, the key molecular mechanisms involved in the female–male interaction, the subsequent triggering of female reproductive development, and the main molecular events occurring during this process of differentiation have not been determined. Because female reproduction requires continuous stimulation from the male schistosome, investigating temporal changes in expression profiles in male and female schistosomes during their interplay, especially during the interval spanning pairing to maturation may provide fundamental insights into this mode of reproductive maturation.

Here we determine the time course when *S. japonicum* reaches sexual maturation after the pairing of worms in the mesenteric veins of the mouse. We employ RNA-seq technology to profile the gene expression of females and males at eight time points, and thereby identify genes that likely play a role in pairing and reproduction. The assembled data allow us to propose a hypothesis of male–female interplay whereby stimuli from the male induces the female to synthesize the neuropeptide hormone allatostatin, and for female reproductive development to commence under the control of a hormone similar to insect juvenile hormone (JH) and 20-hydroxyecdysone (20E).

## Results

**S. japonicum pair reaches sexual maturation in 2 weeks.** To obtain *S. japonicum* transcriptomic profiles spanning the entire process of sexual development, we divided worms obtained from infected mice into the following successive developmental stages: pre-pairing, pairing and maturation. By calculating the pairing ratio at these discrete intervals, we confirmed that direct male–female interaction had not taken place before or at 14 days post infection (d.p.i.). However, some pairing had occurred by 16 d.p.i. (Fig. 1a), followed by a rapid increase in pairing whereby

most worms had paired by 22 d.p.i. Mature vitelline cells first appeared in females at 20 d.p.i., $\sim 4$ days after pairing (Fig. 1a). By 24 d.p.i., mature vitelline glands had appeared in most of the females paired with males. Imaging using confocal laser scanning microscopy (CLSM) indicated the onset of morphological changes in the female and male reproductive organs appeared at 14 d.p.i. and continued through 28 d.p.i. (Fig. 1b–d). Before pairing, the testis and ovary were diminutive and barely evident. Following pairing, both the male and female reproductive organs enlarged and differentiated. Mature sperm had appeared in the seminal vesicle of males by 22 d.p.i., at the same time when mature oocytes first formed in females. By 28 d.p.i., both sexes were sexually mature. These *in vivo* time points from 14 to 28 d.p.i. illustrated the spectrum of anatomical changes in *S. japonicum* from nascence to mature sexual differentiation.

**De novo transcriptome analysis reveals novel transcripts.** We sequenced 48 transcriptomes of *S. japonicum* from both sexes at eight time points (14–28 d.p.i.), each with three biological replicates (worms from one infected mouse represented one replicate; Supplementary Figs 1 and 2). Three transcriptomes were assembled using Trinity with pooled RNA-seq reads of both sexes combined, male only or female only, resulting in a total of 20,058, 19,975 and 17,313 transcripts, respectively (Supplementary Fig. 3). These three transcriptomes were merged to generate the *S. japonicum* transcriptome containing 23,099 transcripts with an N50 length of 769 bp. The mean length of these *de novo* transcripts was 481 bp, and $\sim 30\%$ (7,376) were $> 500$ bp in length (Supplementary Fig. 4). The *de novo* *S. japonicum* transcriptome, representing transcripts expressed during the period 14–28 d.p.i. *in vivo,* covered 68.1% of the Sj_V4.0 transcripts and 82.9% of the EST sequences on comparison with available *S. japonicum* databases (Supplementary Data 1). In addition, the *de novo* transcriptome included 4,561 novel transcripts (Supplementary Data 2). These newly identified transcripts provide expression information that enhances annotation of the *S. japonicum* genome.

**Markedly dissimilar gender-specific transcriptomes.** Among the 23,099 transcripts, 20,366 (88.2%) were expressed (RPKM≥1) in both female and male worms (Fig. 2a). The number of transcripts expressed in males at the different time intervals analysed was nearly constant. By contrast, the number of transcripts expressed in females decreased at 26 d.p.i., but increased again at 28 d.p.i. Principal components analysis (PCA) was carried out to obtain a two-dimensional representation of the dataset of 48 RNA-seq profiles. In male worms, the expression profiles of the three biological replicates were close to each other for each time point from 14 to 28 d.p.i. (Fig. 2b), indicating consistency among the biological replicates. In female worms, the RNA-seq data of the three biological replicates at each time point were close to each other from 14 to 20 d.p.i., whereas those from 22 to 28 d.p.i. were scattered. At early, 14–16 d.p.i., time points, the distances between female and male RNA-seq data were minor, suggesting that immature female and male schistosomula exhibited similar expression profiles early in gender development. After pairing and *in vivo* growth, the distances between male and female RNA-seq data increased progressively. Notably, the characteristics of the adult female expression data at 28 d.p.i. were more similar to those of schistosomula than to the expression data of females at 26 d.p.i. As there were numerous eggs *in utero* at 28 d.p.i., we presume that the female RNA-seq data at this time point included information of expressed genes from the eggs, resulting in the changes in expression profile.

On the basis of hierarchical clustering analysis (HCA; Fig. 2c), the expression profiles were clearly clustered into four groups: immature females (14–20 d.p.i.), immature males (14–20 d.p.i.), mature males (22–28 d.p.i.) and mature females (22–28 d.p.i.). Among these groups, the expression profiles of immature females were discrete from those of mature females, and approximately

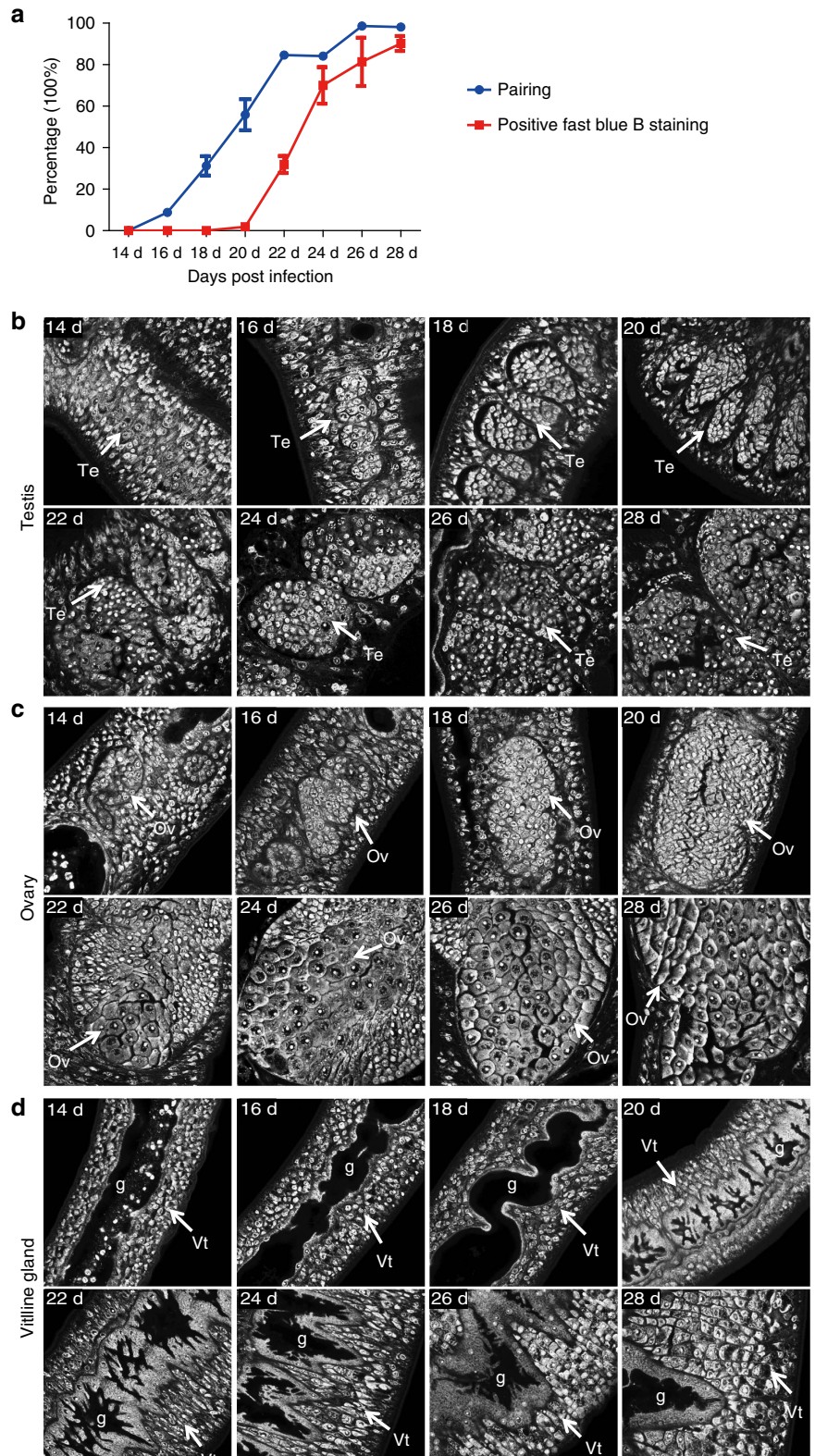

**Figure 1 | *In vivo* male–female pairing and sexual development in *S. japonicum*.** (**a**) Percentage of pairing and vitelline development from 14 to 28 d.p.i. *in vivo*. Positive Fast Blue B staining indicates the mature vitelline cells (mean ± s.e.m., n = 3). (**b–d**) Morphological changes in the reproductive system from 14 to 28 d.p.i., as documented by confocal laser scanning micrographs of the (**b**) testis, (**c**) ovary and (**d**) vitelline gland. g, gut; Ov, ovary; Te, testis; Vt, vitelline cells. n > 5 per experiment. Scale bar, 50 μm.

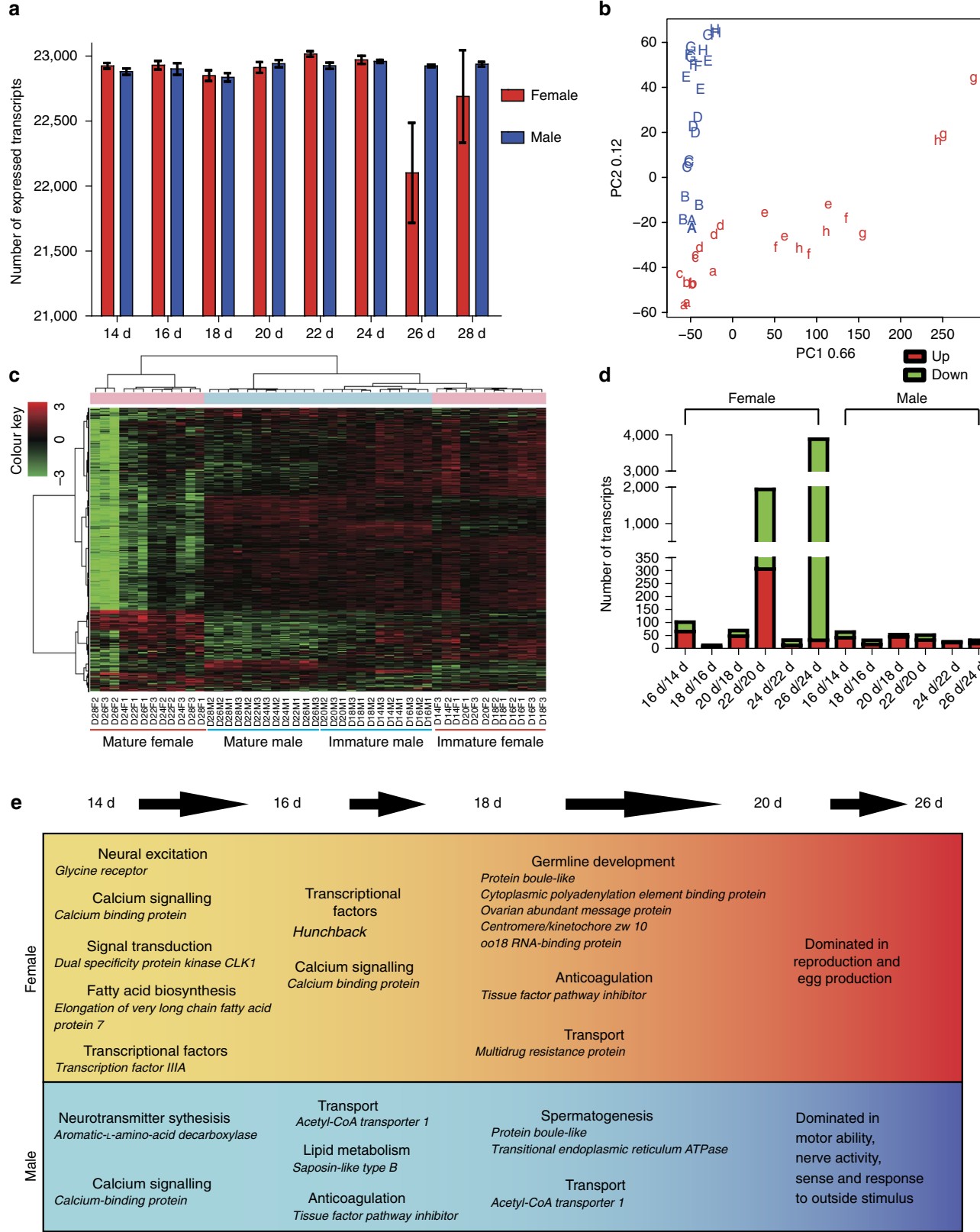

**Figure 2 | Landscape of the transcriptome of *S. japonicum*.** (**a**) Number of expressed genes (RPKM≥1) detected across developmental stages in both males and females (mean ± s.e.m., *n* = 3). (**b**) Principal components analysis (PCA) of a RNA-seq data set of 48 samples. A–H (a–h): 14–28 d.p.i. male (female). (**c**) Hierarchical clustering analysis (HCA) of transcriptional profiles from 48 *S. japonicum* samples with 23,099 genes. F, female; M, male. (**d**) Numbers of differentially expressed transcripts at adjacent time points based on a combination of Student's *t*-test with Bonferroni-corrected *P* < 0.05 and a FC≥2 (or ≤0.5). Up, upregulated based on FC≥2; down, downregulated based on FC≤0.5. (**e**) The enhanced molecular events in male and female worms from day 14 to 26 after infection of the skin by the cercariae.

two-thirds of the female genes were downregulated after sexual maturation. In contrast, the changes between the immature and mature males were minor. Furthermore, the expression profiles of the immature worms of both sexes were clustered into one branch, reflecting a similar gene expression pattern before the emergence of sexual dimorphism. Notably, >50% of the genes of mature male and female schistosomes displayed complementary expression patterns, suggesting that the coupled worm pairs exhibit functional complementation within a sexually reproductive unit.

**Sex-biased transcripts indicate division of labour by 22 d.p.i..** Given the outliers in the RNA-seq data for females at 28 d.p.i., as demonstrated by PCA (Fig. 2b), we used the female and male data from 14 to 26 d.p.i. for further analysis. To explore sex-related transcriptional changes, we compared the temporal expression profiles of the two sexes throughout their interplay. During 14–20 d.p.i., there were ~100 female-biased and ~200 male-biased transcripts at each time point (Supplementary Data 3). With the emergence of mature germ cells at 22 d.p.i. (Fig. 1b–d), the number of differentially expressed transcripts between the two sexes increased markedly. Notably, at 26 d.p.i. when the worms were fully mature, 38.6% (8,907/23,099) of the de novo transcripts were male-dominant, and only 4.2% were female-dominant. Furthermore, gene ontology (GO) analysis revealed functional differences between female and male worms from 22 d.p.i. onwards. For example, at 22 d.p.i., genes involved in the following cellular components and processes were greatly enriched in the female-biased transcripts: ribosome, chromosome, oocyte maturation, cell cycle regulation, transcription, translation, DNA repair, primary metabolic process and folic acid binding (Supplementary Data 4). Genes involved in the following cellular components and processes were enriched in the male-biased transcripts: membrane, cell adhesion, signal transduction, motor activity, muscle fiber development, calcium ion binding, metal ion transmembrane transport, amine transport, aromatic-L-amino-acid decarboxylase activity and synaptic transmission (Supplementary Data 5).

Furthermore, across all seven time points (14–26 d.p.i.), 41 transcripts were always expressed at higher levels in females, while 36 transcripts were always expressed at higher levels in males (Supplementary Figs 5 and 6). Female-specific transcripts included genes functioning in transcriptional regulation, mRNA splicing, DNA repair, germ cell proliferation and cell cycle regulation (Supplementary Data 6). Male-specific transcripts included genes encoding proteins for outer membrane components, transmembrane transport, muscle development, biogenic amine biosynthesis, calcium binding and transcriptional regulation.

**Molecular events occur in different gender from 14 to 26 d.p.i..** We identified 6,535 transcripts in females that were differentially expressed (by ≥2-fold) during sexual development, of which 59 were changed by >100-fold. In males, the expression of 1,934 transcripts changed, but none by ≥64-fold (Supplementary Data 7). As shown in Supplementary Data 8, the 59 transcripts in female worms with considerable (>100-fold) expression changes encoded proteins involved in egg production (eggshell proteins), ovarian maturation (cytoplasmic polyadenylation element binding protein 1, CPEB1), vitelline gland function (extracellular superoxide dismutase, SOD) and cell signalling (G-protein-coupled receptor); all these are associated with reproductive development.

In addition to genes that were differentially expressed at all time points, we found differentially expressed genes by comparing

adjacent time points (Fig. 2d and Supplementary Data 9). In females, two time points had a large number of transcripts with >2-fold changes in expression: at 20–22 d.p.i., 1,671 transcripts were changed (306 upregulated and 1,365 downregulated) and at 24–26 d.p.i., 3,878 transcripts changed (31 upregulated and 3,847 downregulated). In contrast, expression changes between time points in males were modest, where 61 or fewer transcripts differed between any adjacent time points.

Together, these analyses show that the transcriptomic profiles of S. japonicum associated with male–female interactions from unpaired to paired states and sexual maturation represented functional changes in males and females over time. As shown in Fig. 2e and Supplementary Data 9, when pairing began (14–16 d.p.i.), expression increased in males of genes for the nervous system and the production of neurotransmitters, whereas females displayed changes in the expression of genes related to excitation of the nervous system (glycine receptor), fatty acid biosynthesis (elongation of very long fatty acid protein 7), signal transduction (dual specificity protein kinase) and transcriptional regulation (transcription factor IIIA). Moreover, the calcium signalling pathway changed in both sexes. From 16 to 18 d.p.i., genes involved in acetyl-CoA transport and lipid metabolism (saposin-like type B) were upregulated in males. From 18 to 20 d.p.i., genes involved in spermatogenesis (protein boule-like, transitional endoplasmic reticulum ATPase) and the GPCR-mediated neuropeptide signalling pathway (allatostatin-A receptor) were upregulated in males, whereas a cluster of genes related to oocyte maturation and the cell cycle (protein boule-like, CPEB, ovarian abundant message protein and centromere/kinetochore), anticoagulation (tissue factor pathway inhibitor) and acetyl-CoA transport were upregulated in females. After 20 d.p.i., when mature germ cells began to appear, females were committed to reproduction and egg production. Meanwhile, functions subdued by the maturing female worms included motor ability, nerve activity, sense and response to outside stimulus; these functions were supplemented by the male worms.

**Functional genes associated with pairing and reproduction**. To evaluate the time course of expression profiles during the period when S. japonicum undergoes sexual maturation, we performed time series analysis and clustered all 23,099 transcripts into 56 co-expression patterns (Supplementary Fig. 7 and Supplementary Data 10). Using RT–PCR, we confirmed that the correlation of the expression levels of 24 of the transcripts, determined by RNA-seq and quantitative PCR (qPCR), was high (Supplementary Fig. 8). To search for key genes that may participate in the male–female interplay and subsequent reproductive development of female worms, transcripts were identified for which expression patterns exhibited positive correlations with the rates of in vivo male–female pairing and female vitelline development (Fig. 1a). Of the top 100 expressed transcripts that were highly correlated with pairing, 99 were identified in males with only one transcript identified in females (Fig. 3a). Of the male transcripts, 19 were involved in muscle formation and movement regulation (such as calponin-3, myosin essential light chain, titin, troponin I, tropomyosin and dynamin-associated protein), nine were involved in neural development and neurotransmitter transport (such as semaphorin, microtubule-associated protein tau and synaptic vesicular amine transporter), four were involved in extracellular matrix and biological adhesion (such as gynecophoral canal protein, tenascin and annexin), 11 were involved in solute and ion transport (such as AP-1 complex subunit sigma-2, aquaporin-3 and zinc transporter), three encoded tetraspanins (potential vaccine antigens[34]), one encoded an anticoagulant (tissue factor pathway inhibitor) and 24 were uncharacterized (Supplementary

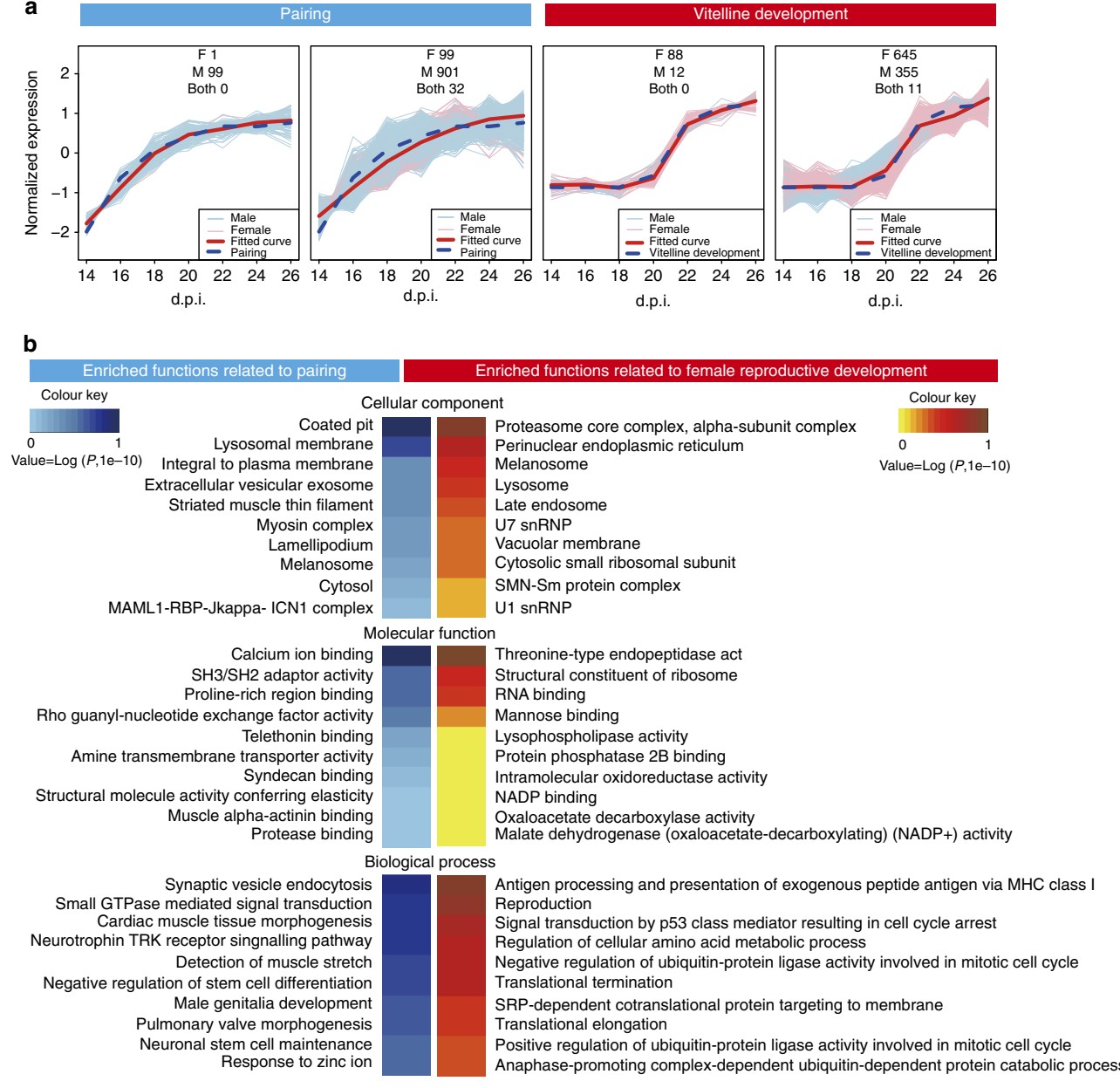

**Figure 3 | Functional genes associated with pairing and reproductive development in schistosomes.** (**a**) Top 100 and 1,000 transcripts in which expression patterns exhibited positive correlations with the rate of *in vivo* male–female pairing and female vitelline development. (**b**) GO enrichment of the genes expressed that were associated with pairing and development of the vitellaria.

Data 11). In addition, of the top 1,000 transcripts, 901 were expressed in males and 99 were in females, with 32 in both sexes (Fig. 3a). Genes involved in the following cellular components and processes were enriched in male transcripts: cell junction, neuronal cell body, amine transmembrane transport, muscle assembly, motor activity, male genitalia development, calcium ion binding, purine/pyrimidine nucleobase transmembrane transport, signalling pathways (neurotrophin TRK, Notch, reelin, MAP kinase, small GTPase, follicle-stimulating hormone stimulus), platelet degranulation, blood vessel endothelial cell differentiation, superoxide metabolic process and positive regulation of nitric oxide synthase activity (Fig. 3b and Supplementary Data 12).

Of the top 100 transcripts expressed that were highly associated with vitelline development, 88 were in females and 12 were in males (Fig. 3a). The female transcripts not only included genes

known to be active in the formation of eggs and vitelline glands (for example, 22 *eggshell proteins*, 2 *tyrosinases*, *major egg antigen*, *ferritin-1* and *SOD*), but also those with roles in the ovarian cell growth, transcriptional regulation, protein synthesis, digestion and host–parasite interaction (Supplementary Data 13). More notable was the presence of a cluster of signal transduction genes, including *GPCR* and *receptor expression-enhancing protein 5*. Of the top 1,000 transcripts, 645 were in female worms and 355 were in males, with 11 transcripts in both sexes (Fig. 3a). The GO analysis showed that the enriched genes of the female transcripts were involved in the following cellular components and processes: reproduction, cell cycle regulation, signal transduction, RNA splicing, ribosome, translation, lipid synthesis and transport, antioxidation, negative regulation of immune response, negative regulation of platelet aggregation and so on (Fig. 3b and Supplementary Data 14).

**Biogenic amine neurotransmitters are linked to pairing.** The RNA-seq data revealed an enhancement of neurologic function in males in conjunction with the pairing process. Transcript *comp34217_c0_seq1* encoded an aromatic-L-amino-acid decarboxylase (AADC) that catalyses the synthesis of biogenic amines such as tryptamine, dopamine and serotonin (Fig. 4a). In animals, biogenic amines are released from synaptic vesicles and mediate a series of physiological functions, such as locomotion, learning and courtship[35–37]. As shown in Fig. 4b, expression of the *Sj AADC* gene continued to rise during the early stages (14–18 d.p.i.) when pairing began and was maintained at a high level in males. Its expression remained low in females. *Sj AADC* was expressed on the internal surface of the male gynecophoral canal, as determined by whole-mount *in situ* hybridization (Fig. 4c). Three synaptic vesicular amine transporters and two sodium-dependent amine transporters also were identified (Fig. 4d); they displayed gradual increase in expression in males, consistent with the increasing production of biogenic amines across all time points. By contrast, the expression of amine transporters increased slightly in females at 14–20 d.p.i., after which expression decreased abruptly. These findings indicated that biogenic amine neurotransmitters play a role in the pairing behaviour of male schistosomes.

**Female maturation involves insect-like hormonal regulation.** The RNA-seq analysis identified a transcript *Fcomp1627_c0_seq1* encoding a typical GPCR that was upregulated >100-fold and had an expression pattern that positively correlated with vitelline development in female schistosomes (Supplementary Data 8 and 13). Further annotation revealed that the protein product was an ortholog of the allatostatin-A receptor, well known in insects. As shown in Fig. 5a, this *SjAlstR* (allatostatin-A receptor-like) gene exhibited distinctive expression patterns between the sexes: in females, *SjAlstR* was expressed at low levels at early stages and increased markedly after 22 d.p.i., whereas in males *SjAlstR* was expressed at low levels in younger worms and slightly increased at later stages. Through whole-mount *in situ* hybridization, we examined the expression of *SjAlstR* in the female gut and ovary (Fig. 5b). Interestingly, *SjAlstR* is also an intestinal receptor in insects[38,39], and its ligand, allatostatin, has been reported in *S. mansoni*[40]. In insects, allatostatin, mediated by AlstR, inhibits the generation of JH, an acyclic sesquiterpenoid that controls insect development and reproduction through its interaction with the molting hormone ecdysone[41]. The RNA-seq transcripts supported a pathway of *de novo* synthesis of the JH precursor (farnesol) from acetyl-CoA (Supplementary Fig. 9). Furthermore, genes for two of four enzymes that participate in the conversion

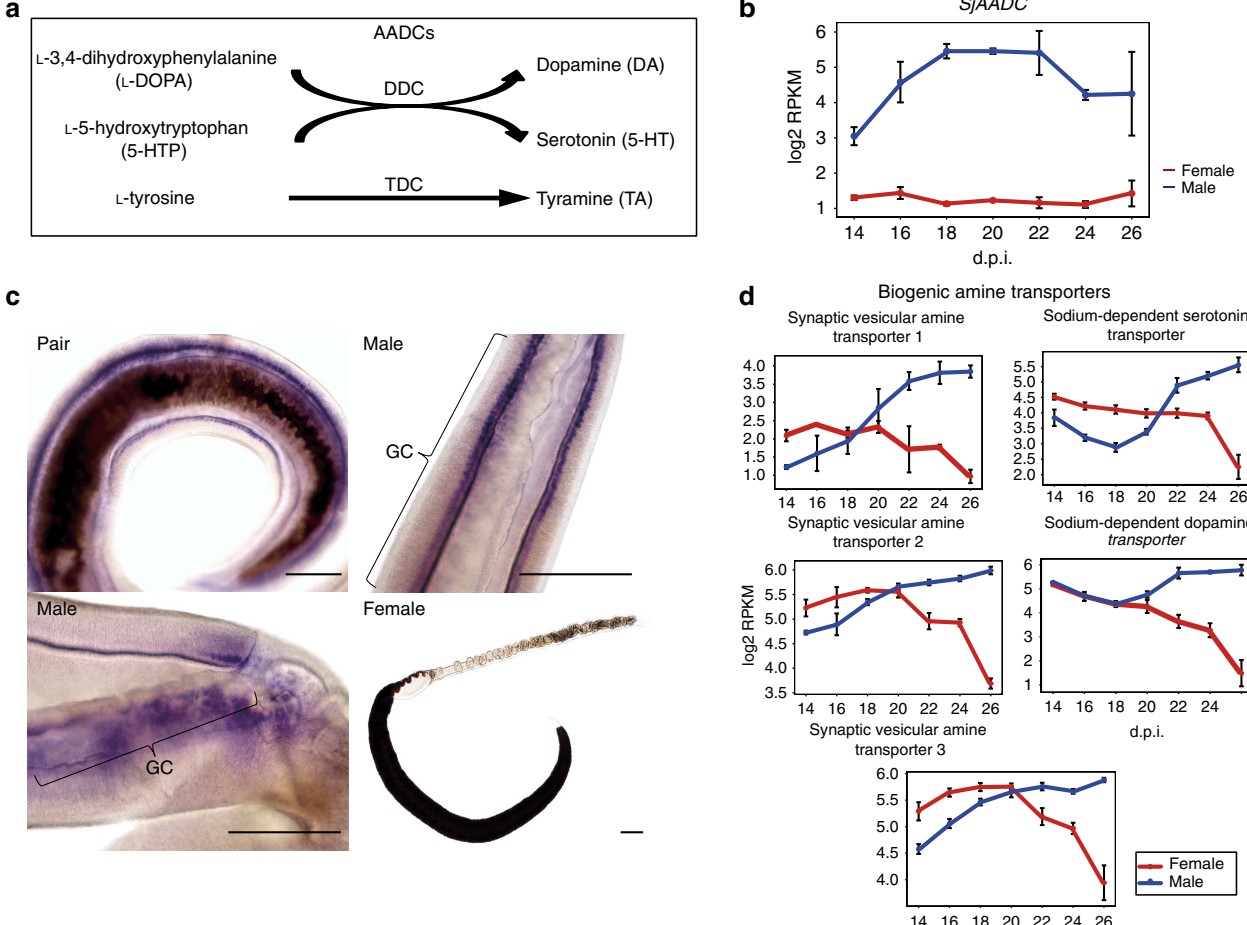

**Figure 4 | Expression of the gene encoding aromatic-L-amino-acid decarboxylase in *S. japonicum* (*Sj*AADC) and amine transporters. (a)** AADC is involved in biosynthesis of biogenic amines. DDC, dopa decarboxylase; TDC, tyrosine decarboxylase. **(b)** The expression pattern of *SjAADC* in both sexes at 14–26 d.p.i. Red, female; blue, male (mean ± s.e.m., n = 3). **(c)** Whole-mount *in situ* hybridization of the *SjAADC* gene. Purple-blue colour indicates positive signals. n > 3 parasites. GC, gynecophoral canal. Scale bars, 500 μm. **(d)** The expression pattern of biogenic amine transporters. Synaptic vesicular amine transporter 1,2,3: >comp5540_c1_seq1 (CAX69443); >comp5301_c0_seq1 (CAX82642); >comp3717_c0_seq1 (AAX26794). Sodium-dependent serotonin transporter (>comp4792_c0_seq2); sodium-dependent dopamine transporter (>comp6056_c0_seq1) (mean ± s.e.m., n = 3).

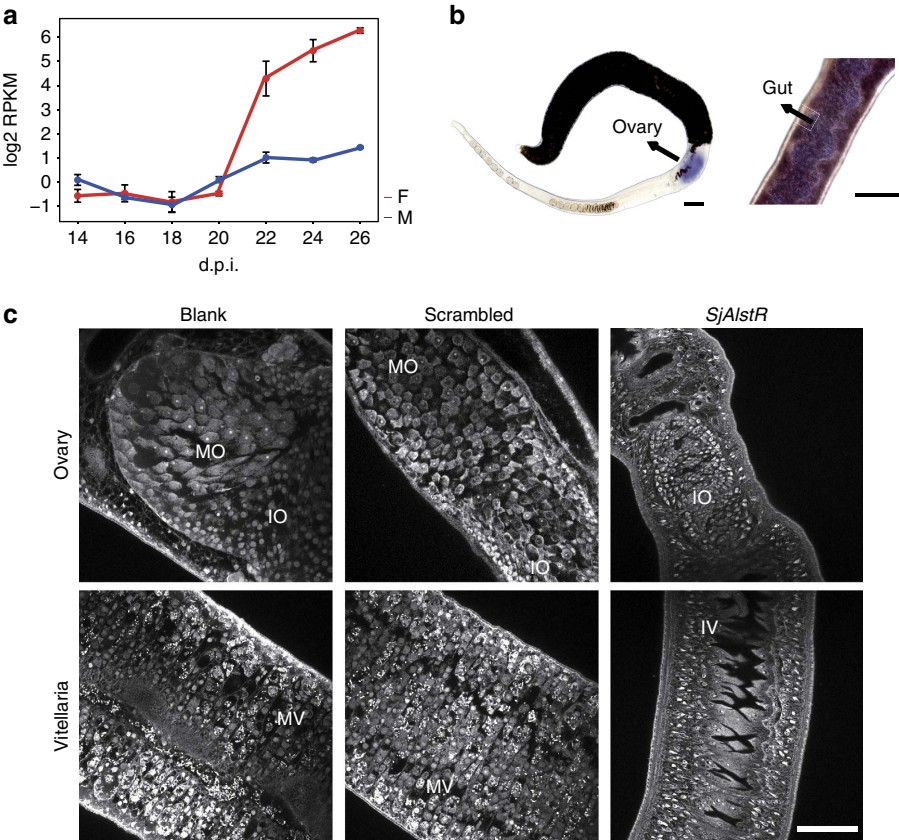

**Figure 5 | Expression and function of *allatostatin-A receptor-like* gene in *S. japonicum* (SjAlstR).** (**a**) The expression pattern of Fcomp1627_c0_seq1 in both sexes at 14–26 d.p.i. Red, female; blue, male (mean ± s.e.m., *n* = 3). (**b**) Whole-mount *in situ* hybridization of the *SjAlstR* gene in the female worm. *n* > 3 parasites. Purplish blue regions indicate presence of the gene product. Scale bars, 100 μm. (**c**) Morphological changes in the female reproductive system after *in vivo* RNAi with blank, scrambled and *SjAlstR* shRNA plasmid. IO, immature oocyte; IV, immature vitelline cell; MO, mature oocyte; MV, mature vitelline cell. *n* > 5 per experiment. Scale bar, 50 μm.

of farnesol to JH were detected in the genome sequences of *S. japonicum* (Supplementary Table 1). In addition, earlier reports had revealed the presence of insect-like ecdysone and a *Drosophila* ecdysone-induced protein 78 ortholog in the genus *Schistosoma*[42–44]. Hence schistosomes appear to possess the same major regulatory elements as insects for the control of reproductive development.

Using RNAi, we silenced the expression of *AlstR in vivo* from 20 d.p.i. when the female commences reproductive development (Supplementary Fig. 10). Ovaries and vitelline glands failed to mature in RNAi-treated females by 23 d.p.i. in contrast to normal development in control females (Fig. 5c). To further determine whether insect hormones influence schistosome physiology, we examined the effect of fenoxycarb (JH analogue)[45] and 20E on schistosomula (18 d.p.i.) and adult worms (28 d.p.i.). At low (1 μg ml$^{-1}$) concentrations of fenoxycarb, the survival and pairing activities of schistosomula and adults *in vitro* were not changed (Fig. 6a). At 10 μg ml$^{-1}$ fenoxycarb, male and female pairing activities were significantly impaired, although the worms were viable. At 100 μg ml$^{-1}$, coupled worms separated within 2 days and died soon thereafter. Unlike fenoxycarb, no effect of 20E was detected on worm activity at the concentrations tested (Supplementary Fig. 11). Using CLSM, we confirmed that the adult female reproductive system was apparently impaired by low concentrations (1 μg ml$^{-1}$) of insect hormones (Fig. 6b). In the fenoxycarb group, mature oocyte cells were seriously damaged, and the structure of mature oocytes was deformed in the 20E group. In contrast, the morphological characteristics of the male

testis did not show any differences between hormone treatment groups and controls (Fig. 6b).

## Discussion

To our knowledge, this is the first study of the dynamic transcriptomic profiles of schistosomes during the period of male–female worm pairing to reproductive development and maturation. We generated 23,099 transcripts by *de novo* assembly, including 4,561 novel transcripts, which provided considerable gene expression information to shed light on the mechanism of male-induced, female sexual development.

Differences occur between the transcriptomes of immature and mature female schistosomes[23–25]. The new data presented here fill in the gap of the continuous transcriptional changes that are evident during these two distinct female reproductive states. At key intervals, 20–22 and 24–26 d.p.i., virgin female schistosomes undergo dramatic changes in gene expression before they complete maturation. These expression changes were highly correlated with morphological changes in sexually maturing females. Mature germ cells emerged between 20 and 22 d.p.i., and mature eggs were produced between 24 and 26 d.p.i. In addition, during female reproductive development, 6,535 transcripts were differentially expressed at least two-fold. The expression changes of such a large number of genes in females must rely on a wide range of transcriptional regulation processes. Indeed, these data indicated that some transcription factors, splicing factors, RNA-binding proteins and microRNA biogenesis proteins that

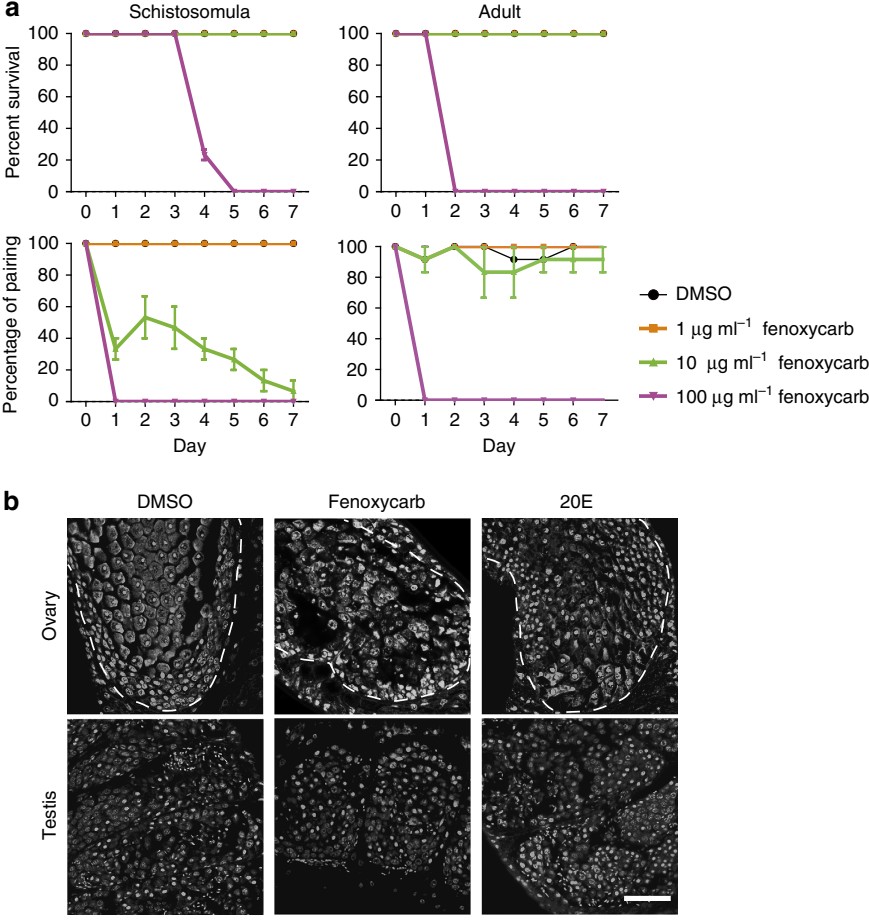

**Figure 6 | Effects of insect hormones on the physical activity and reproduction of *S. japonicum*.** (**a**) Effects of fenoxycarb, an analogue of JH, on the survival and pairing activities of *S. japonicum* (mean ± s.e.m., $n = 3$). (**b**) Confocal laser scanning micrographs documenting morphological changes in the reproductive organs and anatomy of adult *S. japonicum* cultured in $1 \mu g\ ml^{-1}$ fenoxycarb or 20E. $n > 5$ parasites per experiment. Dotted line indicates ovary. Scale bar, 50 μm.

were expressed highly are associated with female reproduction (Supplementary Fig. 12), which merits further studies of their precise functions in the sophisticated regulation of gene expression in schistosomes, especially during female reproductive development.

A division of labour between the adult female and male schistosome is well known as the male controls energy synthesis and the female governs egg production based on gender-related gene expression patterns[28,32]. Our present findings provided much more detail concerning the division of labour by the male and female during their interactions and the reproductive development. Early during the pairing phenomenon (16–20 d.p.i.), the expression profiles of both sexes were similar. Starting from 22 d.p.i., a division of labour was clearly apparent. Specifically, the female worms increased their functions in reproduction, such as the cell cycle, egg formation and protein synthesis, and their intestinal epithelium became more complex, as confirmed by CLSM (Fig. 1d). Functions lost by females, including locomotion, neural activity and response to stimuli, were complemented by male worms. When they were mature, the females—the internal member of the pair of sexually active schistosomes—performed as the super digestive and reproductive 'organs'. Together, the paired worms become a more robust unit than the non-paired worms of either gender, which we predict provides an evolutionary advantage for this remarkable and unconventional reproductive strategy. Notably, GO analysis showed that both males and females enhanced their interplay

with the mammalian host during sexual development. For example, genes involved in the negative regulation of platelet aggregation were enriched in both sexes. Genes related to regulation of the host immune response, such as antigen processing and presentation of exogenous peptide antigen via MHC class I and T-cell-mediated cytotoxicity, were also upregulated. These transcriptional responses indicate that schistosomes use the encoded products of these genes to survive in mammalian host blood by modulating host immune responses. Furthermore, because the host immune system facilitates schistosome development and maturation[46,47], these genes also may participate in these processes.

As the dominant partner in pairing of schistosomes, the male has been thought to support the movement of the female as well as to supply reproductive stimuli. The data here revealed that the male worm reinforced not only the muscular function of the female, but also their neural activity during pairing. Relative to the genes responsible for nervous system functioning, we identified an *AADC*, which catalyses the final step in the biosynthesis of amine neurotransmitters. Biogenic amines, including serotonin, dopamine and histamine, together with their receptors, have been described in *S. mansoni*, and are involved in the neuromuscular signalling that controls motor activity[48–52]. In addition to the control of locomotion, biogenic amines play a key role in the regulation of copulation. In invertebrates, serotonin and dopamine influence and control male mating behaviour[53–55]; in vertebrates, dopamine and

serotonin influence sexual motivation, copulatory motor activity and other behaviours related to sexual activity[35,56]. *AADC* was highly expressed in the gynecophoral canal zone of the *S. japonicum* male. The ortholog in *S. mansoni* is similarly overexpressed in paired adult males compared with non-paired males[57]. These findings indicate strongly that male worms use biogenic amines to control motivation, movement and motility to maintain copulatory activity, a behaviour that is maintained for years. Through the GO analysis of genes with expression patterns that were positively correlated with pairing, genes related to the extracellular region were also found to be enriched, including the gynecophoral canal protein involved in pairing[58], suggesting that these structural molecules are potential tools for males to grasp and hold fast the female.

Our data indicated that insect-like hormonal regulation orchestrates the reproductive development of schistosomes. In insects, the hormones JH and 20E play central roles in development and growth[41]. JH controls aspects of oocyte development, vitellogenin biosynthesis and female receptivity[59]. We detected a *de novo* synthesis pathway of JH precursor (farnesol) from acetyl-CoA (Supplementary Fig. 9) from our RNA-seq data, and further detected two possible genes involved in the first two steps for the conversion of farnesol to JH from the genome data of *S. japonicum* (Supplementary Table 1); ticks exhibit a similar process[60]. However, unlike ticks where JH or JH mimics do not effect development, the JH mimics (farnoxycarb) markedly impaired reproduction in female *S. japonicum*, as in insects. We showed here that the JH analog, fenoxycarb killed schistosomes at high concentration in culture and caused abnormal development of oocytes at lower concentration, in similar manner to insects[45,61]. This finding suggested the presence of a JH or JH-like component in schistosomes, and thus we speculate that schistosomes use a discrete pathway to convert farnesol to JH or a JH-like molecule compared to insects. Although the ecdysone synthesis pathway was not evident with the new transcriptome data, the insect-like 20E and ecdysone receptor have been reported in schistosomes[42–44], suggesting that schistosomes may use an alternative pathway to synthesize ecdysone. In insects, JH concentration is modulated by the neuropeptide allatostatin. Allatostatin-like molecules occur in the nervous system of *S. mansoni* and many other invertebrates[40]. We identified an allatostatin receptor-like molecular with an expression pattern that conformed tightly to vitelline development. This receptor was expressed in the ovary and in the gut, which is proximal to the vitelline gland. Silencing by RNAi inhibited reproductive development in the female. Accordingly, we hypothesize that schistosome allatostatin receptor and its orthologs in insects share a common function as an integrator of feeding and reproduction.

Taken together, we propose a hypothesis to describe the mechanism of male-induced female reproductive development (Supplementary Fig. 13). Male worms develop a strong neuromuscular function together with some extracellular matrix components, such as the gynecophoral canal protein[58], to sense and grasp females in the gynecophoral canal. The male produces the nerve cell-released biogenic amine neurotransmitters by AADC to maintain the pairing behaviour. On receipt by the female of the stimulus, allatostatin is released and transported from the nervous system to the ovary and gut, where it triggers the G-protein-mediated signalling pathway through the allatostatin-A receptor, thereby altering the available concentrations of the schistosome equivalents of JH and 20E. The ovary and vitelline gland mature under the direction of these two hormones, reminiscent of an insect-like regulatory network of development. Since the biogenic amine neurotransmitters and allatostatin are all generated by the nervous system (NS) and

schistosomes have a well-developed NS, the NS likely coordinates pairing activity and reproductive development. However, some aspects remain unclear. First, what stimulus is passed from male to female and how does the female sense it during pairing? Although our analysis of the sex-biased genes did not lead us to predict that biosynthesis pathways in males would generate unusual molecules, we have identified a large number of novel or hypothetical proteins, which we speculate may participate in the process of stimulation. Further studies are now needed to characterize their precise functions. Second, does insect-like JH occur in schistosomes? Third, how does AlstR regulate female reproduction? And, fourth, genes without annotation and noncoding RNAs may provide further insight into the mechanism of male-induced female reproduction.

Our data also revealed potential anti-schistosome drug and vaccine targets. Given that the inflammatory response to entrapped schistosome eggs is the major cause of pathogenesis in schistosomiasis, we recommend that *Sj*AlstR (a GPCR family member) be investigated as an intervention target for this disease. In males, the top 100 transcripts whose expression patterns had positive correlations with rates of *in vivo* pairing included two tetraspanins, which have shown encouraging potential as protective antigens against *S. mansoni*[62,63]. In addition, the male transcripts included other membrane proteins, such as the gynecophoral canal protein, which is a possible target for inhibiting pairing[64]. These surface exposed antigens appear to be involved both in the pairing process and in immune regulation. To date, praziquantel remains the only drug approved to treat schistosomiasis. We observed that the insect growth regulator fenoxycarb showed strong negative effects on *S. japonicum*. Because fenoxycarb has low toxicity in mammals—the oral LD50 for rats is > 16,800 mg kg$^{-1}$ (ref. 65), this drug may offer promise as the basis of a new therapeutic for the neglected tropical disease of schistosomiasis.

To conclude, we mapped the detailed transcriptional changes both in male and female *S. japonicum* that occur from when they begin their interplay until sexual maturation. We now propose a hypothesis for the male-induced female reproduction in schistosomes: insect-like hormonal regulation of sexual maturation in the female schistosome. By enabling a deeper understanding of the reproductive biology of schistosomes, the new findings may promote continued investigation into the evolution of reproduction in these pathogens and into novel approaches for their treatment and control.

## Methods

**Infection of mice with S. japonicum.** *Oncomelania hupensis* snails infected with *S. japonicum* used for the release of the cercariae of *S. japonicum* (Anhui isolate) were provided by the Department of Vector Control of the National Institute of Parasitic Diseases, Chinese Center for Diseases Control and Prevention, Shanghai. Ten-week-old female C57 strain mice (18–20 g), purchased from Shanghai Animal Center, Chinese Academy of Sciences (Shanghai, China) were infected with *S. japonicum* (below). Mice were group-housed (five per cage) and kept in a room with controlled temperature (22 ± 2 °C) and humidity (60–80%) under a 12 h light/dark cycle. All mouse studies were approved by the Ethics Committee of the National Institute of Parasitic Diseases, Chinese Center for Disease Control and Prevention in Shanghai, China (ref. no.: 20100525–1). The use of mice in these experiments was conducted in adherence to the guidelines for the Care and Use of Laboratory Animals of the Ministry of Science and Technology of People's Republic of China ((2006)398). Mice were killed by cervical dislocation before worms were recovered.

**Pairing and reproductive development of schistosomes.** Thirty-two C57 female mice were infected with 80–200 cercariae, each percutaneously through the abdomen. Four mice at each time point (14, 16, 18, 20, 22, 24, 26 and 28 d.p.i.) were killed for recovery of schistosomes. To calculate the pairing status *in vivo*, worms from each mouse (three mice in all) were transferred to DMEM (37 °C), and the number of single female, single male and pairs was determined by examinining under light microscopy. Females were manually separated from males, after which

worms were fixed in 70% ethanol. The developmental status of the vitellaria in females was investigated by staining the worms with Fast Blue B. The number of worms that could pair instead of the total number of worms in each mouse was considered as the total number for statistical analysis. To validate the developmental status of male and female reproductive organs by CLSM, worms recovered from one mouse at each stage were separated by sex and stained with hydrochloric carmine (see below).

**Histochemistry and microscopy.** For Fast Blue B staining, female worms were fixed in 70% ethanol for $\geq 24$ h. After staining with filtered 1% (w/v) Fast Blue B solution, they were dehydrated through an ethanol gradient from 70 to 100%, and then mounted in neutral balsam (Sinopharm Chemical Reagent Co., Ltd, China)[66]. Staining of vitelline cells was evaluated using light microscopy (Nikon NI-SS, Japan). For hydrochloric carmine staining, worms were separated by sex and fixed in AFA (alcohol 95%, formalin 3%, glacial acetic acid 2%). Worms were stained with hydrochloric carmine for 30 min and destained in acidic 70% ethanol. After sequential dehydration in graded ethanol (70, 90, 100%), worms were mounted on glass slides with neutral balsam. Confocal images were taken with a Leica TCS-SP5 Spectral Laser Scanning Confocal Microscope (Leica, Germany) using a 488-nm He/Ne laser.

**Collection of schistosomes for RNA-seq.** To obtain sufficient worms from each mouse for RNA extraction, we infected 24 mice, which allowed inclusion of eight time points *in vivo* with three biological replicates containing different numbers of cercariae, as described in Supplementary Table 2. Worms were recovered by hepatic-portal perfusion at 14, 16, 18, 20, 22, 24, 26 and 28 d.p.i. These worms were transferred into sterile DMEM and separated by sex under light microscopy. The sex of worms was re-checked by microscopic examination, after which the schistosomes were washed three times with sterile DMEM before storage at $-80\,^{\circ}\mathrm{C}$ in RNA later (Qiagen, Germany).

**RNA-seq.** Schistosomes were homogenized in a Bertin Minilys tissue homogenizer (MD, USA), and total RNA was isolated using Qiagen miRNeasy Mini Kit (Valencia, CA). RNA quality was assessed using an Agilent Bioanalyzer; RIN $> 7.5$ for all samples. RNA sequencing was performed according to the manufacturer's protocol using the Illumina TruSeq RNA Sample Preparation Kit and SBS Kit v3 (San Diego, CA). Briefly, 100 ng of each total RNA sample was mixed with 2 µl of 1:1,000 diluted ERCC RNA Spike-in control Mix 1 or Mix 2 (catalogue no. 4456740/4456739, Life Technologies). This was used for polyA mRNA selection and fragmentation, followed by first and second strand synthesis, end repair, adenylation of 3′ ends and adapter ligation. Each library was enriched by 15 cycles of PCR, after which size distribution of products was validated using an Agilent Bioanalyzer and a DNA 1000 Kit. The insert size of the final library was in a band ranging from 200 to 500 bp with a peak at ~260 bp. Libraries were quantified with a Qubit 2.0 Fluorometer (Life Technologies, Grand Island, NY) and sequenced on a HiScanSQ System (Illumina).

**De novo transcriptome assembly and annotation.** First, RNA-seq reads were filtered with Trimmomatic to remove adapter and low-quality sequences. Because the schistosomes were obtained from experimentally infected mice parasites, we aligned the reads to the mouse genome using Bowtie 2 (v0.12.7) and Tophat (v2.0.0). Non-aligned reads were considered to be *S. japonicum*-specific. The female reads, male reads and pooled reads were used for *de novo* transcript assembly in Trinity. Transcripts $\leq 150$ bp in length were discarded. These three transcriptomes were merged to remove redundant sequences with blastn. *S. japonicum* expression databases (V4.0 and EST) were downloaded from http://schistoDB.net/. Finally, *de novo* reconstructed unigenes were annotated by blastx to public databases (the Swiss-Prot protein database, *Caenorhabditis elegans* transcriptome, *Clonorchis sinensis* transcriptome, the UniProt protein database, *S. mansoni* transcriptome, *S. haematobium* transcriptome, *S. japonicum* V4.0 and the NCBI non-redundant (nr) database) with a threshold of $10^{-10}$.

**Analysis of gene expression profiles and gene annotation.** Hierarchical clustering analysis (HCA) was performed with R (www.r-project.org/) using Ward linkage based on a distance matrix of the Pearson correlation of the samples. Clustering was conducted through the hclust function in R using Euclidean distance. Other analyses, including Pearson correlation, Student's *t*-test, PCA and HCA, were performed using functions in R as follows: 'cor', 't.test', 'prcomp' in the 'stats' package and heatmap.2 in the 'gplots' package.

Comparisons among developmental stages of each sex were accomplished by time course, differential expressed gene analysis. To identify time-related genes in each sex, we used a combination of Student's *t*-test with Bonferroni-corrected $P < 0.05$ and a fold change (FC) $\geq 2$ to select genes that were differentially expressed between developmental stages. Sex-related genes were examined between female and male parasites. At any development stage, genes with a FC $\geq 2$ (or $\leq 0.5$) and $P < 0.05$ were considered to be sex-biased, whereas at all eight stages, genes with a FC $\geq 2$ (or $\leq 0.5$) and $P < 0.05$ were considered to be sex-specific.

GO terms were obtained by blast on localized Blast2GO (v2.7.1). The GO enrichment of a cluster of genes was performed through false discovery rate analysis. The pathways of *S. japonicum* were constructed using the online tool KEGG Automatic Annotation Server (KAAS, http://www.genome.jp/tools/kaas/). The assignment of the *de novo* S. japonicum transcripts to KEGG orthologs was implemented with a single-directional best hit method.

**Validation of RNA-seq by quantitative RT–PCR.** Total RNA was reverse transcribed using the High-Capacity cDNA Reverse Transcription Kit (Applied Biosystems, Foster City, CA). Twenty-four transcripts were randomly selected to validate the RNA-seq data, with the *PSMD*[67] gene as a reference. qPCR was performed in a 10-µl reaction containing 1 µl of cDNA, 5 µl of FastStart Universal SYBR Green Master (Roche Applied Science, Indianapolis, IN) and 300 nM each of forward and reverse primers. Gene-specific primers for qPCR were designed using NCBI/Primer-BLAST (http://www.ncbi.nlm.nih.gov/tools/primer-blast). Primer sequences are shown in Supplementary Data 15. The qPCR reactions were performed using the ViiA 7 Real-Time PCR System (Applied Biosystems) as follows: 10 min at 95 °C followed by 40 cycles of 15 s at 95 °C, and 1 min at 60 °C. Specificity was verified by melt-curve analysis. The comparative threshold cycle (Ct) method was used to determine the relative expression levels of target genes. Ct values were averaged and normalized to that of *PSMD*.

**Whole-mount *in situ* hybridization.** Separated male and female parasites were fixed for 20 min in 4% paraformaldehyde dissolved in PBSTX (1X PBS, 0.3% Triton X-100). Fixation of male worms required an additional step—incubation for 10 min at 37 °C in pre-heated (37 °C) reduction solution (50 mM DTT, 1% NP40, 0.5% SDS in 1× PBS). Fixed worms were dehydrated in graded methanol (50%, 100%) and stored at $-20\,^{\circ}\mathrm{C}$ for $>1$ h. Samples were bleached for 24 h in 6% bleach solution (30% $H_2O_2$ diluted in 100% methanol) under direct light. After bleaching, samples were rinsed with 100% $CH_3OH$ and rehydrated by incubation in 50% $CH_3OH$ dissolved in PBSTX followed by incubation in PBSTX. Rehydrated samples were treated with proteinase K solution (2 µg ml$^{-1}$ proteinase K, 0.1% SDS in 1× PBS) for 20 min and re-fixed in 4% paraformaldehyde for 10 min. Hybridization was undertaken at 56 °C and processed as described[68,69]. Digoxigenin (DIG)-labelled probes were synthesized by PCR as follows: 100–200 ng PCR product; 2 µl of 10X DIG RNA Labeling Mix (catalogue no. 11277073910, Roche); 2 µl of 10× transcription buffer; 2 µl T7 RNA polymerase (20 U µl$^{-1}$) and RNase-free water to a final volume of 20 µl. The reaction was incubated for 2 h at 37 °C, after which 2 µl of RNase-free DNase I (10 U µl$^{-1}$) was added and incubated for 10 min at 37 °C. Probes were purified with MEGAclear Kit (catalogue no. AM1908, Ambion) and stored at $-80\,^{\circ}\mathrm{C}$. The primers used to generate PCR products are provided in Supplementary Table 3.

**In vivo RNA interference.** To construct a siRNA expression plasmid (P-422), a 21 nt siRNA sequence (5′-ATGGGCATTAATTGCATGCAT-3′) was designed at Whitehead (http://sirna.wi.mit.edu/) to target the *SjAlstR* gene. On the basis of this 21 nt target sequence, two complementary 55 nt siRNA template oligonucleotides were designed, synthesized, annealed and ligated into P-silencer 4.1-CMV neo plasmid (Ambion, USA). A control, scrambled sequence (5′-GCGAGTACCTTGT TAGATATA-3′) was cloned into P-silencer4.1-CMV neo to serve as a negative control (P-S422).

For each mouse at 20 d.p.i., a 1.5 ml transfection mixture, which contained 0.9% NaCl, 100 µg plasmid DNA and 150 µg PEI (AparnaBio, USA), was injected via tail vein. After maintenance of these mice for the subsequent 72 h, the mice were killed. The females in paired worms were collected for qRT–PCR investigation and for visual examination using light microscopy and CLSM. Mice that were not subjected to the injection of test compounds served as controls. Five independent experiments were performed.

**Evaluation of the effect of insect hormones on worm activity.** To obtain schistosomula (18 d.p.i.), each mouse was infected with ~250–300 cercariae and for adults (28 d.p.i.), each mouse was infected with ~80–100 cercariae. The freshly recovered parasites were washed several times with sterile saline and cultured in RPMI-1640 containing 20% calf serum, 100 IU ml$^{-1}$ penicillin sodium, 100 IU ml$^{-1}$ streptomycin and 0.25 µg ml$^{-1}$ amphotericin B (Hyclone, USA). In cultivation, eight pairs of schistosomula (18 d.p.i.) and four pairs of adults (28 d.p.i.) were placed in each well containing 4 ml culture medium, as above. Fenoxycarb and 20E (34343, H5142, Sigma Aldrich, USA), dissolved in dimethylsulphoxide, were added to obtain the final working concentrations of 1, 10 and 100 µg ml$^{-1}$. Control worms were cultured in medium supplemented with an equal volume of dimethylsulphoxide. Media and hormones were replenished every 2–3 days. Plates were incubated for 7 days at 37 °C under 5% $CO_2$ in air. Pairing and survival progress of the worms were monitored daily by light microscopical examination of the cultures.

**Data availability.** All raw data and the *de novo* assembled transcript sequences were submitted to GenBank with the project accession number of PRJNA343582. Nucleotide sequences have been deposited in the Sequence Read Archive (SRA) of

NCBI under accession codes SRS1704021 (Sj-male-1), SRS1705432 (Sj-male-2); SRS1705606 (Sj-male-3); SRS1705608 (Sj-male-4); SRS1708578 (Sj-male-5); SRS1708583 (Sj-male-6); SRS1708585 (Sj-male-7); SRS1708587 (Sj-male-8); SRS1708590 (Sj-male-9); SRS1708592 (Sj-male-10); SRS1708594 (Sj-male-11); SRS1708596 (Sj-male-12); SRS1710531 (Sj-male-13); SRS1710593 (Sj-male-14); SRS1710600 (Sj-male-15); SRS1710602 (Sj-male-16); SRS1710605 (Sj-male-17); SRS1714235 (Sj-male-18); SRS1714237 (Sj-male-19); SRS1714239 (Sj-male-20); SRS1714241 (Sj-male-21); SRS1714243 (Sj-male-22); SRS1714245 (Sj-male-23); SRS1714247 (Sj-male-24); SRS1703988 (Sj-female-1); SRS1705431 (Sj-female-2); SRS1705599 (Sj-female-3); SRS1705607 (Sj-female-4); SRS1708574 (Sj-female-5); SRS1708582 (Sj-female-6); SRS1708584 (Sj-female-7); SRS1708586 (Sj-female-8); SRS1708588 (Sj-female-9); SRS1708591 (Sj-female-10); SRS1708593 (Sj-female-11); SRS1708595 (Sj-female-12); SRS1710496 (Sj-female-13); SRS1710591 (Sj-female-14); SRS1710598 ( Sj-female-15); SRS1710601 (Sj-female-16); SRS1710604 (Sj-female-17); SRS1714234 (Sj-female-18); SRS1714236 (Sj-female-19); SRS1714238 (Sj-female-20); SRS1714240 (Sj-female-21); SRS1714242 (Sj-female-22); SRS1714244 (Sj-female-23); SRS1714246 (Sj-female-24). See also Supplementary Table 4 for information on samples. All other relevant data are available from the authors upon request.

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

## Acknowledgements

We thank Li Jin, Guoping Zhao, Hong Ma, Shimin Zhao, Ting Ni, Yufang Zheng, Sheng Li and Jiekun Xuan for the comments on the manuscript. We also thank Chao Guo and Charles Wang for conducting RNA extraction and RNA-seq library construction through a service contract. This work was supported by the National Natural Science Foundation of China (No. 81271867 and 91431104), International Science and Technology Cooperation Program of China (No. 2014DFA31130) and National Science & Technology Major Program (No. 2009ZX10004-302).

## Author contributions

W.H., L.S. and J.W. conceived the project. J.W., S.W., N.L. X.M. and B.X. carried out the infections and sample collection. H.S. performed the *de novo* assembly of *S. japonicum* transcriptome. Y.Y., H.S., T.Q. and J.W. analysed the RNA-seq data. J.W., M.L. and R.C. contributed the morphological data acquisition. Y.Z. carried out the RT–PCR experiments. J.W. and Q.L. contributed the RNAi experiments. J.W., Y.Y., Y.Z., M.S., P.J.B., D.P.M., Z.F., L.S. and W.H. contributed to the writing and revision of the manuscript. All authors reviewed and approved the submitted manuscript.

## Additional information

**Competing financial interests:** The authors declare no competing financial interests.

