## [Peer Review File · Nature Communications]

Reviewers' comments:

Reviewer #1 (Remarks to the Author):

The interplay between male and female schistosomes has long been of interest to many scientists in the field. What makes it of general interest is that the male schistosome send a signal independent of sperm transfer that regulates female-specific gene expression and this expression leads to reproductive development of the female schistosome. What makes this significant is that this enables the female to produce eggs which are responsible for all the pathogenesis of schistosomiasis and transmission to the snail intermediate host. Therefore contributions such as the present one that contribute to our understanding of this interplay are significant. This study defined the time course when *S. japonicum* reaches sexual maturation after the pairing of worms and coupled this with transcriptome analysis. Even though development is asynchronous in the mouse, they chose time points where they could observe major changes. This process lead to the identification of over 5000 novel transcripts that will be useful to the field for further studies. A nice comparison of male and female-specific transcription as well as overall male and female transcription is presented during development. Somewhat surprising of the top 100 transcripts associated with pairing 99 were identified in males and only 1 in females. This ratio more or less held for the top 1000 transcripts. Out of this study two genes were identified, a gene that catalyzes the synthesis of biogenic amines that are hypothesized to play a role in male pairing behavior and a GPCR that correlated positively with vitelline development in female schistosomes. It is the vitelline genes that are regulated by the male signal.

Overall this in depth molecular approach to understanding pairing and subsequent female reproductive development of *Schistosoma japonicum* is very important. Most of the previous work has been done on *Schistosoma mansoni*. This will set the stage for a comparison between *S. japonicum* and *S. mansoni* and *haematobium*.

Minor Comments:

Overall the paper is well written. There are two suggestions:

1. Line 59, schistosomes are not monogamous and therefore pairing is not permanent but it is continuous- recommend continuous pairing
2. Line 68; insight not sight

Reviewer #2 (Remarks to the Author):

This paper by Wang et al describes an analysis of gene expression patterns associated with male/female pairing and reproductive development of *Schistosoma japonicum*.

Transcriptome profiling was done by RNAseq at different times post-infection, starting before the coupling of males and females, all the way to day 28, when the worms are fully paired and sexually mature. RNAseq data were further validated by a combination of quantitative RT-PCR, in situ hybridization, RNAi and pharmacological studies. The results describe several gender-biased transcripts, including novel transcripts that are likely to play

an important role in schistosome coupling and sexual development. On the whole, this is an interesting, novel study. Although there have been transcriptome studies of this type in the related species, *S. mansoni*, this is a first for *S. japonicum* and it is a more detailed, comprehensive analysis than what is currently available for any species of schistosome.

There are a few weaknesses. As a general comment, I find the paper suffers from too much speculation about the mechanism(s) by which males induce female development. The model in Fig. 6 illustrates the problem. While the evidence suggests (possibly) the involvement of biogenic amines and allatostatin, there is no direct evidence linking these systems in any of the proposed steps. In particular, the link between the allatostatin pathway and JH production by the females should be tested experimentally since it is central to the entire model. I acknowledge these additional experiments are substantial and perhaps beyond the scope of the study. Nevertheless, without additional evidence, the model is premature and should be either removed, or at least made less speculative.

Major comments:

Line 290: The notion that males stimulate female sexual maturation via biogenic amine signaling is based largely on one transcript (AY812557.1), which is significantly up-regulated in paired male schistosomes. The transcript is described as an aromatic amino acid decarboxylase (AADC) but a search on NCBI shows it to be a homologue of tyrosine decarboxylase (*tdc-1*). AADC and *tdc-1* are actually different enzymes. AADC catalyzes the second step in serotonin and dopamine biosynthesis, whereas *tdc-1* catalyzes the synthesis of tyramine and (indirectly) octopamine. *Tdc-1* (Smp_135230) is also up-regulated in paired adult male *S. mansoni* (see ref 51). These findings could suggest involvement of tyramine and/or octopamine transmitters but not serotonin or dopamine. This should be revised throughout. AADC should be changed to *tdc-1* in the text and all the figures.

A related concern is that the putative *tdc-1* localizes entirely to the gynecophoric canal (Fig. 4). This is unusual for a neuronal enzyme; one would expect to see expression in the nervous system, where the transmitters are synthesized. It is also surprising that no other genes associated with biogenic amine signaling (e.g. receptors, other biosynthetic enzymes) are up-regulated. It is possible the *tdc-1* homologue is working in a different capacity, which is unrelated to neuronal signaling. To be clear, I am not suggesting that the biogenic amine model is necessarily wrong but it needs to be revised and alternative explanations should be considered.

Line 301 and Fig. 4 - Can they predict what types of "amine transporters" are up-regulated in the males? Are they serotonin, dopamine or octopamine? Are they vesicular or plasma membrane transporters (or both)? Vesicular transporters are proton-dependent, not ATP-dependent as stated (Fig. 4D). Also, the schematic in Fig 4A should be revised to describe the reactions catalyzed by AADC and *tdc-1* (see above).

Line 330 and Fig. 5C - The RNAi results must be validated at the RNA level by qPCR. It is unclear if this was done. Also they should provide some information on the reproducibility of the RNAi - How many worms were tested? What proportion of females showed the RNAi phenotype (Fig. 5C)?

Line 334 and Fig. 5D, 5E - The results with fenoxycarb are hard to follow. They state that 10 µg/ml drug causes significant loss of male-female pairing but this does not show on the graph (Fig 5D, adults, lower panel). Also, the drug effect on the female reproductive system is hard to see (Fig. 5E).

Supplemental Tables S5, S7 and others: Given that so many researchers work with *S. mansoni*, it would be helpful if they could show the *S. mansoni* orthologues for each of these genes of interest (using the *smp* designation).

Other comments:

line 64- ...transferred to the female, which (insert comma)

line 66- "...mechanisms associated with male-induced female development have not been identified to date." - While it is true that the precise mechanisms are unclear, there has been some important work in this area, notably from the LoVerde and Grevelding labs. This work should be cited here.

line 238: "Meanwhile, functions lost by the maturing female worms included motor ability, nerve activity, sense and response to outside stimulus." - What is the evidence that motor ability or nerve activity were "lost" in the females? The sentence is misleading because it conveys the impression that females have diminished motor function compared to males. Yet, adult (mature) females are as active as males in culture. This needs to be clarified.

line 246 quantitative RT-PCR

line 366 - highly expressed

line 494 - vertebrate

Reviewer #3 (Remarks to the Author):

The manuscript describes the use of high-throughput RNA-seq to identify the large-scale gene expression changes of male and female *Schistosoma japonicum* during early development in the vertebrate host, across 8 time points throughout the sexual developmental process from pairing to maturation. Clear sex-related expression patterns were demonstrated, which revealed unambiguous functional division between males and females during their maturation. The approach is novel and interesting. Identification of a gene that encoded an AADC (Aromatic-L-amino-acid decarboxylase) that catalyzes the synthesis of biogenic amines such as tryptamine, dopamine and serotonin was accompanied by the observation that expression of the *Sj* AADC gene continued to rise during the early stages (14-18 dpi) when pairing began and was maintained at a high level in males, whereas expression of this gene remained low in females. In addition, amine transporters were identified as expressed in males and females, and whole-mount *in situ* hybridization showed that *Sj* AADC was expressed on the internal surface of the male gynecophoral canal. The results suggest that biogenic amine neurotransmitters may play a role in the pairing behavior of male schistosomes. In addition, the work identified transcripts encoding all of the enzymes that make up the mevalonate pathway from acetyl-CoA to farnesal, and the authors speculate that an insect-like hormonal regulatory mechanism involving Juvenile

Hormone (JH) may act in the reproductive development of females, although no genes encoding the enzymes that convert farnesal to JH were found. In general, the experiments were well designed, the data is of good quality, the RNA-seq assembly and the statistical analyses of differential expression were adequately performed. I have some major concerns regarding the presentation of data, which are described in detail below.

Major points:

1) My major concern regarding the presentation of the data is the lack of cross annotation between the de novo assembled transcripts reported here and the publicly available annotated Sj genes. There are over 27600 mRNA sequences for *S. japonicum* in GenBank with gene name annotations, most of them derived from the Liu F. et al. Nature 2009 genome paper. The authors should give the corresponding accession number and the gene name for each of their transcripts that match 69.1% of the Sj_V4.0 transcripts, as reported on line 127 of the manuscript, and these accession numbers should be shown in the main text and in the Tables. Also, it is very important for the authors to deposit their raw RNA-seq data in a public repository such as SRA, as well as for them to make their de novo assembly available at the NCBI Transcriptome Shotgun Assembly (TSA) database. As it is now, the readers can have no detailed information at all, regarding any of the transcripts mentioned in the text. No sequence is available for any transcript; no degree of identity and/or similarity is given between the Sj transcript sequence and the proposed gene homolog in other species, throughout the text.

2) The authors indicate on line 129-130 that the de novo transcriptome included 5,205 novel transcripts, and they cite "Supplementary Table 1". No novel transcripts are mentioned in Supplementary Table 1. This table only has a summary of the numbers of Sj_V4.0 and EST transcripts publicly available, and the number of public genes with matches in the de novo transcriptome. It is very important that the authors identify these 5,205 novel transcripts among the transcripts deposited in the TSA record. I also suggest that the authors look for conserved protein domains possibly encoded by these novel transcripts, by running a blast search against the PFAM database, and that they annotate the novel transcripts that show matches to conserved domains. Instead of just indicating "novel transcript" as the gene name for every one of these genes, an indication of their partial similarity to known conserved domains (when available) could shed additional light on their possible function.

3) A general problem that I found throughout the manuscript can be illustrated in the presentation of the results for female-biased and male-biased transcripts during the 14 to 20 days post infection, on lines 171-173. The authors mention that they identified "~100 female-biased and ~200 male-biased transcripts at each time point (Supplementary Table 2)". When going to Supplementary Table 2, I find only the number of genes at each time point, and no further information. Which were the genes, their names? The authors mention some of the GO enriched processes to which these genes belong, however the readers can not explore this information in the future, because no gene names, no gene sequences, no hard data is provided. This problem appears throughout the text, and I think that the authors need to add detailed gene names information for all the results described. In

addition, the gene accessions for the transcripts comprising each enriched GO should be given in supplementary excel tables (transcript number and/or Sj accession number). Also, I suggest that the authors provide the list of all 23,099 transcripts in an excel table, and that they indicate for example, in one column next to each gene, into which of the 56 co-expression patterns the gene was clustered. The illustration on Supplementary Fig 7 gives an overview of the results, however for the interested reader the data will only be useful if the authors give a label number to each of the 56 clusters in Supplementary Fig 7, and indicate this cluster number next to each of their 23,099 transcripts. This excel file could contain many columns that would show a number of other features that were identified for a given transcript, such as male-biased (x time point) transcript, male specific transcript, belongs to pathway x, etc.

4) On line 247-249 the authors stated: "Using RT-PCR, we confirmed that the correlation of the expression levels of 24 of the transcripts, determined by RNA-seq and qPCR, was high (Supplementary Fig 8)." No statistical significance for the mentioned correlation was calculated. The authors should provide some metrics for the RNA-seq versus qPCR correlation comparison.

5) On lines 310-314 the authors describe the "identification of a transcript Fcomp1627_c0_seq1 encoding a typical GPCR [...] (Supplementary Table 7 and Supplementary Table 10). [...] Further annotation revealed that the protein product was an ortholog of the allatostatin-A receptor, well-known in insects." Upon inspection of Supplementary Table 7 and Supplementary Table 10 I found that Fcomp1627_c0_seq1 is annotated as "Cephalotocin receptor" and not as allatostatin-A receptor. The authors should give the details about the identity and similarity of Fcomp1627_c0_seq1 compared with Cephalotocin receptor and allatostatin-A receptor. They subsequently named Fcomp1627_c0_seq1 as SjAlstR (allatostatin-A receptor-like) gene, and it is important that the data that permitted this annotation be well described, with accession numbers for the allatostatin-A receptor and Cephalotocin receptor from the other species.

6) Most important is the incomplete/incorrect description of the findings related to the claim of identification of the insect juvenile hormone (JH) pathway. Thus, on lines 323-328 the authors stated: "The RNA-seq transcripts revealed the existence of a pathway of de novo synthesis of an equivalent of JH from acetyl-CoA (Supplementary Fig 9). [...] It seems that schistosomes possess the same major regulatory elements as insects for the control of reproductive development." The authors found the genes encoding all of the enzymes that make up the mevalonate pathway from acetyl-CoA to farnesal, however they omitted from this description the fact that they found no genes encoding the enzymes that in insects convert farnesal to JH, such as the farnesal dehydrogenase, the JH methyltransferase, the JH epoxidase. Although the authors have used fenoxycarb, a JH analog, and found that the hormone affected male-female pairing and worm viability, I find it premature and inappropriate for the authors to speculate that an insect-like hormonal regulatory mechanism involving Juvenile Hormone (JH) may act in the reproductive development of females. In particular, I call the authors' attention to the fact that other invertebrate arthropods such as ticks and other acari have the same mevalonate complete pathway from acetyl-CoA to farnesal, however, different from insects, they appear not to make JH but

rather only a JH precursor (see Zu, J. et al. (2016) PLoS ONE, <http://dx.doi.org/10.1371/journal.pone.0141084>). The authors should change the Results, the Discussion (lines 421-422), the Abstract and the final part of their Introduction, to make it clear that no transcripts encoding any of the enzymes that convert farnesal to JH were detected, and they should discuss alternative possibilities.

Minor points:

7) On lines 263 - 264 the text states: "of the top 1,000 transcripts, 901 were expressed in males and 99 were in females, with 32 in both sexes (Fig 3a)." The numbers given in the panels inside Fig. 3a are different from the ones in the main text. The same problem appears with the main text on line 283 versus Fig. 3a.

8) On lines 300-302 the authors state: "In addition, amine transporters were identified (Fig 4d) that displayed a gradual increase in expression in males, [...]". This is part of the same problem that I already mentioned above, but I think that it is worth repeating here: the three amine transporter transcripts are identified only by their de novo contig names, only in Fig 4d panel, and no specific information is given regarding the gene names, the percent similarity to gene orthologs, the accession numbers of the orthologs, etc. And yet the transport of amines is an important part of the results highlighted by the work.

9) On lines 342 and 346, Fig 5d should read Fig 5e.

10) In the Introduction on lines 73-80 the authors call the attention to the fact that: "RNA-seq is an insightful, formidable transcriptomics tool capable of delivery of genome-scale transcription profiles unconstrained by genomic annotation (refs 13-17), in contrast to microarrays-based analysis. Transcriptomic studies by microarray or RNA-seq have been performed with mature and immature females, and with different sexes in *Schistosoma japonicum* and *S. mansoni*. Compared with immature females, the genes involved in egg production and hemoglobin digestion are highly enriched in adult females (refs 18-22); Compared with adult females, the expression of genes associated with tegument structure and movement are enriched in adult males (refs 23-27)." In spite of citing the two papers that used RNA-seq to study gender associated gene expression in *S. japonicum*, the authors failed to cite the paper by Anderson et al. (2015) PLoS NTD, DOI:10.1371/journal.pntd.0004334, where RNA-seq was used to analyze gender associated gene expression in *S. mansoni*.

11) On lines 453-457 the authors note that "the analysis of the sex-biased genes did not lead us to predict that biosynthesis pathways in males would generate special molecules" and because of this they "speculate that the tactile interaction between the pair may provide the pivotal stimulus from the male worm etc. [...]." I would like to note that a large number of male-biased genes were "novel transcripts" or genes encoding "hypothetical proteins", and I believe that before these genes are characterized, it is inappropriate to make a statement of lack of generation of "special molecules".

Reviewers' comments:

Reviewer #1 (Remarks to the Author):

The interplay between male and female schistosomes has long been of interest to many scientists in the field. What makes it of general interest is that the male schistosome send a signal independent of sperm transfer that regulates female-specific gene expression and this expression leads to reproductive development of the female schistosome. What makes this significant is that this enables the female to produce eggs which are responsible for all the pathogenesis of schistosomiasis and transmission to the snail intermediate host. Therefore contributions such as the present one that contribute to our understanding of this interplay are significant. This study defined the time course when *S. japonicum* reaches sexual maturation after the pairing of worms and coupled this with transcriptome analysis. Even though development is asynchronous in the mouse, they chose time points where they could observe major changes. This process lead to the identification of over 5000 novel transcripts that will be useful to the field for further studies. A nice comparison of male and female-specific transcription as well as overall male and female transcription is presented during development. Somewhat surprising of the top 100 transcripts associated with pairing 99 were identified in males and only 1 in females. This ratio more or less held for the top 1000 transcripts. Out of this study two genes were identified, a gene that catalyzes the synthesis of biogenic amines that are hypothesized to play a role in male pairing behavior and a GPCR that correlated positively with vitelline development in female schistosomes. It is the vitelline genes that are regulated by the male signal.

Overall this in depth molecular approach to understanding pairing and subsequent female reproductive development of *Schistosoma japonicum* is very important. Most of the previous work has been done on *Schistosoma mansoni*. This will set the stage for a comparison between *S. japonicum* and *S. mansoni* and *haematobium*.

Minor Comments:

Overall the paper is well written. There are two suggestions:

1. Line 59, schistosomes are not monogamous and therefore pairing is not permanent but it is continuous- recommend continuous pairing
2. Line 68; insight not sight

Response:

We appreciate the reviewer's positive assessment on the importance of this report. In the revised manuscript we have modified the text according to both of the reviewer's comments.

Reviewer #2 (Remarks to the Author):

This paper by Wang et al describes an analysis of gene expression patterns associated with male/female pairing and reproductive development of *Schistosoma japonicum*.

Transcriptome profiling was done by RNAseq at different times post-infection, starting before the coupling of males and females, all the way to day 28, when the worms are fully paired and sexually mature. RNAseq data were further validated by a combination of quantitative RT-PCR, in situ hybridization, RNAi and pharmacological studies. The results describe several gender-biased transcripts, including novel transcripts that are likely to play an important role in schistosome coupling and sexual development. On the whole, this is an interesting, novel study. Although there have been transcriptome studies of this type in the related species, *S. mansoni*, this is a first for *S. japonicum* and it is a more detailed, comprehensive analysis than what is currently available for any species of schistosome.

There are a few weaknesses. As a general comment, I find the paper suffers from too much speculation about the mechanism(s) by which males induce female development. The model in Fig. 6 illustrates the problem. While the evidence suggests (possibly) the involvement of biogenic amines and allatostatin, there is no direct evidence linking these systems in any of the proposed steps. In particular, the link between the allatostatin pathway and JH production by the females should be tested experimentally since it is central to the entire model. I acknowledge these additional experiments are substantial and perhaps beyond the scope of the study. Nevertheless, without additional evidence, the model is premature and should be either removed, or at least made less speculative.

Response:

We greatly appreciate the reviewer's positive assessment on the novelty and comprehensiveness of our work.

We agree with the comments about the level of speculation on the molecular mechanism of male-induced female reproduction depicted in Fig. 6. Therefore, instead of calling it a model, we now refer it as a hypothesis. In addition, we have relocated the original Fig. 6 to the supplementary information (see Supplementary Fig. 13). We anticipate that revision adequately addresses the reviewer's concern.

Major comments:

Line 290: The notion that males stimulate female sexual maturation via biogenic amine signaling is based largely on one transcript (AY812557.1), which is significantly up-regulated in paired male schistosomes. The transcript is described as an aromatic amino acid decarboxylase (AADC) but a search on NCBI shows it to be a homologue of tyrosine decarboxylase (tdc-1). AADC and tdc-1 are actually different enzymes. AADC catalyzes the second step in serotonin and dopamine biosynthesis, whereas tdc-1 catalyzes the synthesis of tyramine and (indirectly) octopamine. Tdc-1 (Smp_135230) is also up-regulated in paired adult male *S. mansoni* (see ref 51). These findings could suggest involvement of tyramine and/or octopamine transmitters but not serotonin or dopamine. This should be revised throughout. AADC should be changed to tdc-1 in the text and all the figures.

Response:

AADCs are homodimeric pyridoxal 5'-phosphate (PLP) enzymes that can decarboxylate naturally occurring L-aromatic amino acids [1]. Both tyrosine decarboxylase (Tdc) and dopa decarboxylase (Ddc) belong to AADCs but the difference between them is that Ddc decarboxylates L-dopa and 5-hydroxytryptophan whereas Tdc decarboxylates tyrosine [2,3].

We performed Blastp on Swiss-prot using putative *S. japonicum* AADC (AY812557.1); it shared high similarity to well-annotated TDC in *C. elegans* (43% identity, 130E-150) and aromatic-L-amino-acid decarboxylase in *H. sapiens* (44.4% identity, 920E-141).

Alkema et al [2] obtained deletion alleles of five putative AADC genes in C. elegans and eventually named the deleted gene responsible for the absence of tyramine/octopamine as the tyrosine decarboxylase. So far, sequence analysis of the orthologous S. japonicum transcript revealed a typical AADCs domain, the pyridoxal-dependent decarboxylase conserved domain. However, whether it decarboxylates tyrosine or L-dopamine remains to be determined. Experiments like those performed in C. elegans may be needed to define its function before the S. japonicum 'orthologue' can be named. Therefore the more generic term AADC seems the appropriate term at present.

- 1. Zhu, M.Y. & Juorio, A.V. Aromatic L-amino acid decarboxylase: biological characterization and functional role. *Gen Pharmacol* 26, 681-96 (1995).**
- 2. Alkema, M.J., Hunter-Ensor, M., Ringstad, N. & Horvitz, H.R. Tyramine Functions independently of octopamine in the *Caenorhabditis elegans* nervous system. *Neuron* 46, 247-60 (2005).**
- 3. Livingstone, M.S. & Tempel, B.L. Genetic dissection of monoamine neurotransmitter synthesis in *Drosophila*. *Nature* 303, 67-70 (1983).**

A related concern is that the putative *tdc-1* localizes entirely to the gynecophoric canal (Fig. 4). This is unusual for a neuronal enzyme; one would expect to see expression in the nervous system, where the transmitters are synthesized. It is also surprising that no other genes associated with biogenic amine signaling (e.g. receptors, other biosynthetic enzymes) are up-regulated. It is possible the *tdc-1* homologue is working in a different capacity, which is unrelated to neuronal signaling. To be clear, I am not suggesting that the biogenic amine model is necessarily wrong but it needs to be revised and alternative explanations should be considered.

Response:

Although AADCs play key roles in the nervous system, expression of AADC is not restricted to the nervous system. TDC-1 has been observed in neurons and in gonadal sheath cells in *C. elegans* [1]. In *D. melanogaster*, the dopa decarboxylase gene (*Ddc*) is expressed in the hypoderm and the nervous system [2]; and in mammals, AADC mRNA is also detected in extraneuronal tissues [3,4]. On the other hand, schistosomes have a rich nerve plexus that extends along the body surface including the tegument [5]. In addition, in schistosomes both the dopamine receptor and the serotonin receptor are expressed in the peripheral innervation of the muscles of the body wall muscles and in the tegument [6,7]. In situ hybridizations confirmed that *Sj* AADC was expressed along the gynecophoric canal, and may be expressed in the nervous system (two strong staining lines along the body, Fig. 4c). Because of the technical limitations and sensitivity of this method, however, we could not determine definitively whether it is expressed in the neuronal or extraneuronal region.

- 1. Alkema, M.J., Hunter-Ensor, M., Ringstad, N. & Horvitz, H.R. Tyramine Functions independently of octopamine in the *Caenorhabditis elegans* nervous system. *Neuron* 46, 247-60 (2005).**
- 2. Morgan, B.A., Johnson, W.A. & Hirsh, J. Regulated splicing produces different forms of dopa decarboxylase in the central nervous system and hypoderm of *Drosophila melanogaster*. *EMBO J* 5, 3335-42 (1986).**
- 3. Blechingberg, J., Holm, I.E., Johansen, M.G., Borglum, A.D. & Nielsen, A.L. Aromatic l-amino acid decarboxylase expression profiling and isoform detection in the developing porcine brain. *Brain Res* 1308, 1-13 (2010).**
- 4. Eaton, M.J. et al. Distribution of aromatic L-amino acid decarboxylase mRNA in mouse brain by in situ hybridization histology. *J Comp Neurol* 337, 640-54 (1993).**

5. de Saram, P.S. *et al.* Functional mapping of protein kinase A reveals its importance in adult *Schistosoma mansoni* motor activity. *PLoS Negl Trop Dis* 7, e1988 (2013).
6. El-Shehabi, F., Taman, A., Moali, L.S., El-Sakkary, N. & Ribeiro, P. A novel G protein-coupled receptor of *Schistosoma mansoni* (SmGPR-3) is activated by dopamine and is widely expressed in the nervous system. *PLoS Negl Trop Dis* 6, e1523 (2012).
7. Patocka, N., Sharma, N., Rashid, M. & Ribeiro, P. Serotonin signaling in *Schistosoma mansoni*: a serotonin-activated G protein-coupled receptor controls parasite movement. *PLoS Pathog* 10, e1003878 (2014).

Line 301 and Fig. 4 - Can they predict what types of "amine transporters" are up-regulated in the males? Are they serotonin, dopamine or octopamine? Are they vesicular or plasma membrane transporters (or both)? Vesicular transporters are proton-dependent, not ATP-dependent as stated (Fig. 4D). Also, the schematic in Fig 4A should be revised to describe the reactions catalyzed by AADC and tdc-1 (see above).

Response:

Based on Blast results, these three genes showed high similarity to mammalian synaptic vesicular amine transporters, which function in the transport of biogenic amine neurotransmitters, including dopamine, norepinephrine, serotonin, and histamine [1-3]. In addition, we identified a sodium-dependent serotonin transporter (>comp4792_c0_seq2) and a sodium-dependent dopamine transporter (>comp6056_c0_seq1), according to Ribeiro et al. [4]. Both showed similar expression patterns with the synaptic vesicular amine transporters (Fig. 4d). On the other hand, we could not detect in our de novo transcriptome the homologous sequence of the octopamine transporter (Smp_193800) mentioned by Ribeiro et al. [4].

We have modified the decarboxylase reaction (see Fig. 4a) based on the reports by Alkema et al. and Livingstone et al. [5,6], as well as increasing the description of the amine transports expression as “Three synaptic vesicular amine transporters and two sodium-dependent amine transporters were also identified” in Line 292-294.

- 1. Howell, M. et al. Cloning and functional expression of a tetrabenazine sensitive vesicular monoamine transporter from bovine chromaffin granules. *FEBS Lett* 338, 16-22 (1994).**
- 2. Erickson, J.D. & Eiden, L.E. Functional identification and molecular cloning of a human brain vesicle monoamine transporter. *J Neurochem* 61, 2314-7 (1993).**
- 3. Thiriot, D.S., Sievert, M.K. & Ruoho, A.E. Identification of human vesicle monoamine transporter (VMAT2) luminal cysteines that form an intramolecular disulfide bond. *Biochemistry* 41, 6346-53 (2002).**
- 4. Ribeiro, P. & Patocka, N. Neurotransmitter transporters in schistosomes: structure, function and prospects for drug discovery. *Parasitol Int* 62, 629-38 (2013).**
- 5. Alkema, M.J., Hunter-Ensor, M., Ringstad, N. & Horvitz, H.R. Tyramine Functions independently of octopamine in the *Caenorhabditis elegans* nervous system. *Neuron* 46, 247-60 (2005).**
- 6. Livingstone, M.S. & Tempel, B.L. Genetic dissection of monoamine neurotransmitter synthesis in *Drosophila*. *Nature* 303, 67-70 (1983).**

Line 330 and Fig. 5C - The RNAi results must be validated at the RNA level by qPCR. It is unclear if this was done. Also they should provide some information on the reproducibility of the RNAi - How many worms were tested? What proportion of females showed the RNAi phenotype (Fig. 5C)?

Response:

Q-RT-PCRs were performed to validate the RNAi results. In five independent mice (5-8 worms/mouse), the expression level of the *SjAlstR* gene was reduced by ~40-60% after treatment with *shRNA* plasmid. Sexually developed females were not found in the *sh-SjAlstR* treatment group, whereas in the control group 33-50% worms showed sexual development (see Supplementary Fig. 10 and its fig legend).

Line 334 and Fig. 5D, 5E - The results with fenoxycarb are hard to follow. They state that 10 $\mu\text{g/ml}$ drug causes significant loss of male-female pairing but this does not show on the graph (Fig 5D, adults, lower panel). Also, the drug effect on the female reproductive system is hard to see (Fig. 5E).

Response:

In Fig. 5d, the third bar on each day shows the male-female pairing status in the 10 $\mu\text{g/ml}$ drug treatment groups. In fact, there should be four bars (DMSO, 1 $\mu\text{g/ml}$, 10 $\mu\text{g/ml}$ and 100 $\mu\text{g/ml}$) on each day. Because the percentage of the pairing under 100 $\mu\text{g/ml}$ drug was 0 on each day, no visible bars are shown on the graph. Now we added the data of day 0 on each chart of Fig .5d.

To clarify drug effect on female reproduction, we have annotated the micrograph with a dotted line in the region of the ovary; and have included a revised version of Fig. 5 of higher resolution.

Supplemental Tables S5, S7 and others: Given that so many researchers work with *S. mansoni*, it would be helpful if they could show the *S. mansoni* orthologues for each of these genes of interest (using the smp designation).

Response:

We concur this would be informative. We now provide information for orthologues from both *S. mansoni* and *S. haematobium* in Supplementary Tables 2, 6, 8, 11 and 13.

Other comments:

line 64-transferred to the female, which (insert comma)

line 66- "...mechanisms associated with male-induced female development have not been identified to date." - While it is true that the precise mechanisms are unclear, there has been some important work in this area, notably from the LoVerde and Grevelding labs. This work should be cited here.

Response:

We apologize for the oversight. Certainly we agree that these groups have contributed pioneering knowledge to this field. We have now cited relevant studies from the Grevelding and Loverde labs (line 63).

line 238: "Meanwhile, functions lost by the maturing female worms included motor ability, nerve activity, sense and response to outside stimulus." - What is the evidence that motor ability or nerve activity were "lost" in the females? The sentence is misleading because it conveys the impression that females have diminished motor function compared to males. Yet, adult (mature) females are as active as males in culture. This needs to be clarified.

Response:

The dynamic transcriptomic profiles revealed that the expression levels of the genes related to motor ability and neurological activity were similar or identical in both genders at the early stages, whereas expression levels were continuously reduced in

females and were significantly lower than those in males after maturation, indicative that the mature female has weaker motor/neural functions than the male. We concur that it is inappropriate to use the term “lost”; accordingly, we have revised the description to “subdued”.

line 246 quantitative RT-PCR

line 366 - highly expressed

line 494 - vertebrate

Thank you; corrected as suggested.

Reviewer #3 (Remarks to the Author):

The manuscript describes the use of high-throughput RNA-seq to identify the large-scale gene expression changes of male and female *Schistosoma japonicum* during early development in the vertebrate host, across 8 time points throughout the sexual developmental process from pairing to maturation. Clear sex-related expression patterns were demonstrated, which revealed unambiguous functional division between males and females during their maturation. The approach is novel and interesting. Identification of a gene that encoded an AADC (Aromatic-L-amino-acid decarboxylase) that catalyzes the synthesis of biogenic amines such as tryptamine, dopamine and serotonin was accompanied by the observation that expression of the S_j AADC gene continued to rise during the early stages (14-18 dpi) when pairing began and was maintained at a high level in males, whereas expression of this gene remained low in females. In addition, amine transporters were identified as expressed in males and females, and whole-mount in situ hybridization showed that S_j AADC was expressed on the internal surface of the male gynecophoral canal. The results suggest that biogenic amine neurotransmitters may play a role in the pairing behavior of male schistosomes. In addition, the work identified transcripts encoding all of the enzymes that make up the mevalonate pathway from acetyl-CoA to farnesal, and the authors speculate that an insect-like hormonal regulatory mechanism involving Juvenile Hormone (JH) may act in the reproductive development of females, although no genes encoding the enzymes that convert farnesal to JH were found. In general, the experiments were well designed, the data is of good quality, the RNA-seq assembly and the statistical analyses of differential expression were adequately performed. I have some major concerns regarding the presentation of data, which are described in detail below.

Response:

We appreciate the reviewer's positive assessment on our work and the detailed, constructive comments. We have carefully and fully addressed these comments in the revised manuscript.

Major points:

1) My major concern regarding the presentation of the data is the lack of cross annotation between the de novo assembled transcripts reported here and the publicly available annotated Sj genes. There are over 27600 mRNA sequences for *S. japonicum* in GenBank with gene name annotations, most of them derived from the Liu F. et al. Nature 2009 genome paper. The authors should give the corresponding accession number and the gene name for each of their transcripts that match 69.1% of the Sj_V4.0 transcripts, as reported on line 127 of the manuscript, and these accession numbers should be shown in the main text and in the Tables. Also, it is very important for the authors to deposit their raw RNA-seq data in a public repository such as SRA, as well as for them to make their de novo assembly available at the NCBI Transcriptome Shotgun Assembly (TSA) database. As it is now, the readers can have no detailed information at all, regarding any of the transcripts mentioned in the text. No sequence is available for any transcript; no degree of identity and/or similarity is given between the Sj transcript sequence and the proposed gene homolog in other species, throughout the text.

Response:

We concur. In the R1 version, we have provided the cross annotation between the de novo assembled transcripts and the publicly available annotated genes of S. japonicum. By cutting the sequence similarity by over 75%, 82.9% (6,678/8,067) of the Sj_ESTs and 68.1% (9,177/13,469) Sj_V4.0 transcripts matched our de novo transcriptome (see Table S1).

We have submitted the raw RNA-seq data at DDBJ/EMBL/GenBank under the accession GEZP00000000 (Project ID PRJNA343582). The version described in this paper is the first version, GEZP01000000. It is under the manually review process and

will be released after that. It is our sincere hope that our endeavors as presented here will benefit the research community and to improvement in the public health.

In Table S2, we provided the cross annotation of all 23,099 de novo assembled transcripts with well known databases, such as S. mansoni transcriptome, S. haematobium transcriptome, NR (2016), Uniprot and InterPro Scan.

2) The authors indicate on line 129-130 that the de novo transcriptome included 5,205 novel transcripts, and they cite "Supplementary Table 1". No novel transcripts are mentioned in Supplementary Table 1. This table only has a summary of the numbers of Sj_V4.0 and EST transcripts publicly available, and the number of public genes with matches in the de novo transcriptome. It is very important that the authors identify these 5,205 novel transcripts among the transcripts deposited in the TSA record. I also suggest that the authors look for conserved protein domains possibly encoded by these novel transcripts, by running a blast search against the PFAM database, and that they annotate the novel transcripts that show matches to conserved domains. Instead of just indicating "novel transcript" as the gene name for every one of these genes, an indication of their partial similarity to known conserved domains (when available) could shed additional light on their possible function.

Response:

We appreciate this suggestion; and rechecked our data accordingly. To identify novel transcripts, we performed a blast search and found 18,538 of 23,099 de novo transcripts matched the known Sj_EST or Sj_v4.0 transcripts based on the criteria of E value $< E^5$. To further annotate the remaining 4,561 novel transcripts, we ran the blast search to identify homologues of Sha transcripts (297), Sm transcripts (103), NR2016 (734), Uniprot (4), Blast2GO (75) and InterPro Scan (12). This yielded 3,336 novel transcripts without annotation across all these databases (See Supplementary Table 2).

3) A general problem that I found throughout the manuscript can be illustrated in the presentation of the results for female-biased and male-biased transcripts during the 14 to 20 days post infection, on lines 171-173. The authors mention that they identified "~100 female-biased and ~200 male-biased transcripts at each time point (Supplementary Table 2)". When going to Supplementary Table 2, I find only the number of genes at each time point, and no further information. Which were the genes, their names? The authors mention some of the GO enriched processes to which these genes belong, however the readers can not explore this information in the future, because no gene names, no gene sequences, no hard data is provided. This problem appears throughout the text, and I think that the authors need to add detailed gene names information for all the results described. In addition, the gene accessions for the transcripts comprising each enriched GO should be given in supplementary excel tables (transcript number and/or S_j accession number). Also, I suggest that the authors provide the list of all 23,099 transcripts in an excel table, and that they indicate for example, in one column next to each gene, into which of the 56 co-expression patterns the gene was clustered. The illustration on Supplementary Fig 7 gives an overview of the results, however for the interested reader the data will only be useful if the authors give a label number to each of the 56 clusters in Supplementary Fig 7, and indicate this cluster number next to each of their 23,099 transcripts. This excel file could contain many columns that would show a number of other features that were identified for a given transcript, such as male-biased (x time point) transcript, male specific transcript, belongs to pathway x, etc.

Response:

For the female-biased and male-biased transcripts, we provide detailed information in Table S3. For the Tables of GO enrichment, we have added the list of transcripts following each of the enriched processes (see Supplementary Tables 4, 5, 12 and 14). Additionally, we have added the transcript list of all these 56 clustered groups in Supplementary Table 10. Additional information for the de novo transcripts, such as

the related pathway (GO term), sex-biased feature, is included in Supplementary Tables 2 and 3.

4) On line 247-249 the authors stated: "Using RT-PCR, we confirmed that the correlation of the expression levels of 24 of the transcripts, determined by RNA-seq and qPCR, was high (Supplementary Fig 8)." No statistical significance for the mentioned correlation was calculated. The authors should provide some metrics for the RNA-seq versus qPCR correlation comparison.

Response:

This is correct. We have added the correlation values in Supplementary Fig 8.

5) On lines 310-314 the authors describe the "identification of a transcript Fcomp1627_c0_seq1 encoding a typical GPCR [...] (Supplementary Table 7 and Supplementary Table 10). [...] Further annotation revealed that the protein product was an ortholog of the allatostatin-A receptor, well-known in insects." Upon inspection of Supplementary Table 7 and Supplementary Table 10 I found that Fcomp1627_c0_seq1 is annotated as "Cephalotocin receptor" and not as allatostatin-A receptor. The authors should give the details about the identity and similarity of Fcomp1627_c0_seq1 compared with Cephalotocin receptor and allatostatin-A receptor. They subsequently named Fcomp1627_c0_seq1 as SjAlstR (allatostatin-A receptor-like) gene, and it is numbers for the allatostatin-A receptor and Cephalotocin receptor from the other species.

Response:

Fcomp1627_c0_seq1 (AAW26428.1) showed high sequence similarity to Sha_104441 in S. haematobium, which is annotated as Cephalotocin receptor 1 (See Table 1 below). However, sequence alignment with other species showed that Fcomp1627_c0_seq was more like an allatostatin-A receptor (Table 1). In addition,

both contain a TM7_1 domain (7 transmembrane receptor, rhodopsin family). Because the ligand for this *S. japonicum* GPCR remains to be determined, based on sequence similarity we have termed it *allatostatin-A receptor-like gene*.

Table 1. The similarity of *S. japonicum* *allatostatin-A receptor-like gene* with homologues in other organisms.

Organism	Annotation/ Accession number	E-value/ Identity
Schistosoma haematobium	Cephalotocin receptor 1, putative Sha_104441	4.71E-127
Schistosoma mansoni	Putative peptide (Allatostatin/somatostatin)-like receptor Smp_041880	0.0 80.6%
Drosophila melanogaster	Allatostatin A receptor 2, isoform A NP_001247352.1	1e-15 22%
Bombyx mori	allatostatin-A receptor NP_001037035.1	7e-11 22%
Crassostrea gigas	Cephalotocin receptor 1 EKC38821.1	2e-10 30%
Octopus vulgaris	cephalotocin receptor 1 BAD67169.1	8e-10 25%

6) Most important is the incomplete/incorrect description of the findings related to the claim of identification of the insect juvenile hormone (JH) pathway. Thus, on lines 323-328 the authors stated: "The RNA-seq transcripts revealed the existence of a pathway of de novo synthesis of an equivalent of JH from acetyl-CoA (Supplementary Fig 9). [...] It seems that schistosomes possess the same major regulatory elements as insects for the control of reproductive development." The authors found the genes encoding all of the enzymes that make up the mevalonate pathway from acetyl-CoA to farnesal, however they omitted from this description the fact that they found no genes encoding the enzymes that in insects convert farnesal to JH, such as the farnesal dehydrogenase, the JH methyltransferase, the JH epoxidase. Although the authors have used fenoxycarb, a JH analog, and found that the hormone affected male-female pairing and worm viability, I find it premature and inappropriate for the authors to speculate that an insect-like hormonal regulatory mechanism involving Juvenile Hormone (JH) may act in the reproductive development of females. In

particular, I call the authors' attention to the fact that other invertebrate arthropods such as ticks and other acari have the same mevalonate complete pathway from acetyl-CoA to farnesal, however, different from insects, they appear not to make JH but rather only a JH precursor (see Zu, J. et al. (2016) PLoS ONE, <http://dx.doi.org/10.1371/journal.pone.0141084>). The authors should change the Results, the Discussion (lines 421-422), the Abstract and the final part of their Introduction, to make it clear that no transcripts encoding any of the enzymes that convert farnesal to JH were detected, and they should discuss alternative possibilities.

Response:

We appreciate this comment. According to Zu, J. et al. 2016 PLoS ONE, we performed BLASTp on the S. japonicum database in GeneDB (www.genedb.org/homepage/Sjaponicum) using the proteins involved in insects for JH conversion from farnesal. We identified schistosome proteins with similarity to insect farnesal dehydrogenase and farnesol oxidase, but not JH methyltransferase or JH expodiase (see Table 2 below). Based on current data, S. japonicum, like ticks, has the enzymes to complete the mevalonate pathway from acetyl-CoA to farnesal, but does not have all the enzymes as those present and conserved in insects for conversion of the JH precursor to JH.

However, unlike in ticks where JH or JH mimics did not effect development, JH mimics (farnoxycarb) markedly impacted female reproduction of S. japonicum (as in insects). This suggests the presence in schistosomes of a JH or JH-like component, and thus we speculate that schistosomes use a discrete pathway to convert farnesal to JH or a JH-like molecule compared to insects.

After comprehensive consideration, we decided to maintain the description in Abstract and introduction, but modified the related text in Results in Line 315–318 as “The RNA-seq transcripts revealed the existence of a pathway of de novo

*synthesis of JH precursor (farnesal) from acetyl-CoA (Supplementary Fig. 9), although only two enzymes among four members involving the process of converting farnesal to JH were detected (Supplementary Table 15)”; and Discussion in Line 412-422 as “ We detected a de novo synthesis pathway of JH precursor (farnesal) from acetyl-CoA (Supplementary Fig. 9), but not all the enzymes involving the process of transferring farnesal to JH (Supplementary table 15); which is similar to process in ticks⁶⁰. However, unlike ticks where JH or JH mimics do not effect development, JH mimics (farnoxycarb) markedly impacted reproduction in female *S. japonicum*, as in insects. We showed here that the JH analog, fenoxy carb killed schistosomes at high concentration in culture and caused abnormal development of oocytes at lower concentration, in similar fashion to insects^{46,61}. This finding suggested the presence of a JH or JH-like component in schistosomes, and thus we speculate that schistosomes use a discrete pathway to convert farnesal to JH or a JH-like molecule compared to insects”.*

Table 2. The *S. japonicum* transcripts that matched to genes involved in JH branch in insects

Enzyme	Accession# insects	Accession# S. japonicum	Identity/ e-value
Farnesal dehydrogenase	XP_011492954 A. aegypti	Sjp_0058160.1..pep	29%, 2e-12
JH methyltransferase		NA	NA
Farnesol Oxidase	XP_001606362.2 D. ananassae	Sjp_0036830.1..pep	38%, 1.4e-47
JH epoxidase		NA	NA

Minor points:

7) On lines 263 - 264 the text states: "of the top 1,000 transcripts, 901 were expressed in males and 99 were in females, with 32 in both sexes (Fig 3a)." The numbers given in the panels inside Fig. 3a are different from the ones in the main text. The same problem appears with the main text on line 283 versus Fig. 3a.

Response:

We agree that the way we originally displayed the number in Fig. 3a caused this confusion. For example, the number "901" in the main text is the sum of the number of M (869) and the number of Both (32). We have now changed the numbers on Fig.3a in keeping with those in the main text.

8) On lines 300-302 the authors state: "In addition, amine transporters were identified (Fig 4d) that displayed a gradual increase in expression in males, [...]". This is part of the same problem that I already mentioned above, but I think that it is worth repeating here: the three amine transporter transcripts are identified only by their de novo contig names, only in Fig 4d panel, and no specific information is given regarding the gene names, the percent similarity to gene orthologs, the accession numbers of the orthologs, etc. And yet the transport of amines is an important part of the results highlighted by the work.

Response:

Here we provide the details of these three genes. They show high similarity to mammalian synaptic vesicular amine transporters (see Table 3 below), which function in the transport of biogenic amine neurotransmitters, including dopamine, norepinephrine, serotonin and histamine [1-3]. In addition, we have now included accession number and annotation for each gene in Fig. 4d and the figure legend.

Table 3. The *de novo* transcript of *S. japonicum* that matched to mammalian amine transporter.

De novo transcript Accession# S. japonicum	Accession# Mammals	Annotation in Mammals	Identity/ e-value
>comp5540_c1_seq1 CAX69443	NP_777078.1 Bos taurus	Synaptic vesicular amine transporter	50.0% 160e-150
	NP_766111.1 Mus musculus	Synaptic vesicular amine transporter	49.6% 7.3e-147
	NP_003045.2 Homo sapiens	Synaptic vesicular amine transporter	48.5% 13e-147
	NP_003045.2 Homo sapiens	Synaptic vesicular amine transporter	51.3% 1.7e-153
>comp5301_c0_seq1 CAX82642	NP_777078.1 Bos taurus	Synaptic vesicular amine transporter	50.4% 7.6e-153
	NP_766111.1 Mus musculus	Synaptic vesicular amine transporter	59.9% 1.2e-54
>comp3717_c0_seq1 AAX26794	NP_003045.2 Homo sapiens	Synaptic vesicular amine transporter	59.9% 1.6e-54

1. **Howell, M. et al. Cloning and functional expression of a tetrabenazine sensitive vesicular monoamine transporter from bovine chromaffin granules. *FEBS Lett* 338, 16-22 (1994).**
2. **Erickson, J.D. & Eiden, L.E. Functional identification and molecular cloning of a human brain vesicle monoamine transporter. *J Neurochem* 61, 2314-7 (1993).**
3. **Thiriot, D.S., Sievert, M.K. & Ruoho, A.E. Identification of human vesicle monoamine transporter (VMAT2) luminal cysteines that form an intramolecular disulfide bond. *Biochemistry* 41, 6346-53 (2002).**

9) On lines 342 and 346, Fig 5d should read Fig 5e.

Response:

We agree and have corrected this in the R1 version.

10) In the Introduction on lines 73-80 the authors call the attention to the fact that: "RNA-seq is an insightful, formidable transcriptomics tool capable of delivery of genome-scale

transcription profiles unconstrained by genomic annotation (refs 13-17), in contrast to microarrays-based analysis. Transcriptomic studies by microarray or RNA-seq have been performed with mature and immature females, and with different sexes in *Schistosoma japonicum* and *S. mansoni*. Compared with immature females, the genes involved in egg production and hemoglobin digestion are highly enriched in adult females (refs 18-22); Compared with adult females, the expression of genes associated with tegument structure and movement are enriched in adult males (refs 23-27)." In spite of citing the two papers that used RNA-seq to study gender associated gene expression in *S. japonicum*, the authors failed to cite the paper by Anderson et al. (2015) PLoS NTD, DOI:10.1371/journal.pntd.0004334, where RNA-seq was used to analyze gender associated gene expression in *S. mansoni*.

Response:

We thank the reviewer and have now included citation of this paper in the main text (Line 78).

11) On lines 453-457 the authors note that "the analysis of the sex-biased genes did not lead us to predict that biosynthesis pathways in males would generate special molecules" and because of this they "speculate that the tactile interaction between the pair may provide the pivotal stimulus from the male worm etc. [...]." I would like to note that a large number of male-biased genes were "novel transcripts" or genes encoding "hypothetical proteins", and I believe that before these genes are characterized, it is inappropriate to make a statement of lack of generation of "special molecules".

Response:

We agree with the reviewer's comment and have redrafted the text as: 'Although our analysis of the sex-biased genes did not lead us to predict that biosynthesis pathways in males would generate unusual molecules, we have identified a large

number of novel or hypothetical proteins, which we speculate may participate in the process of stimulation. Further studies are now needed to characterize their precise functions'. (Lines 448-452)

Reviewers' comments:

Reviewer #2 (Remarks to the Author):

This is a re-submission of a paper by Wang et al describing gene expression patterns associated with reproduction in *Schistosoma japonicum*. The authors have addressed my previous concerns and made appropriate revisions. The revised manuscript will make an important, novel contribution to the field.

Reviewer #3 (Remarks to the Author):

In the revised version of the manuscript the authors have adequately addressed most of the questions and problems raised by the referees. Nevertheless, I still have two points that I believe need further attention.

1) I believe that the text describing the two possible enzymes in the JH pathway that the authors now mention in Supplementary Table S15 and in lines 317-318 still does not give a correct description of their findings.

First, the sentence on lines 315-318 start saying that "The RNAseq transcripts revealed a pathway of de novo synthesis of the JH precursor [...]" and continues to mention the finding of two possible enzymes in the last four steps of the pathway, citing Suppl. Table S15 with two genes that are annotated in the *S. japonicum* genome (Sjp_0058160.1..pep and Sjp_0036830.1..pep) without making it clear that there were no RNA-seq transcript reads from the present work matching these two genes predicted by the v4.0 genome (I could not find any denovo assembly entry in Suppl. Table S2 that was annotated as being similar to Sjp_0058160 or to Sjp_0036830).

Second and most important, the enzyme mentioned in Suppl. Table S15 (XP_011492954 *A. aegypti*) as being the Farnesal dehydrogenase is not the correct one. The two Farnesal dehydrogenase genes from *A. aegypti* have been identified in 2013 by Rivera-Perez et al. (doi: 10.1016/j.ibmb.2013.04.002) as being Acc numbers AGI96742 and AGI96740, and I searched by BLASTP the protein database of *S. japonicum* for the presence of genes similar to those two Farnesal dehydrogenase genes with absolutely no success. There seems to be no gene similar to Farnesal dehydrogenase in the annotated *S. japonicum* genome. Also, there seems to be no denovo assembled transcripts for Farnesal dehydrogenase among the RNA-seq data of the authors, since I could not find any denovo transcript in Suppl. Table S2 that has been annotated as similar to AGI96742 and AGI96740 or to AAEL012161, AAEL012162 and AAEL012165, the original *A. aegypti* entries for these enzymes in the insect database.

Also, regarding the entry for the supposed second insect gene in the JH pathway, given in Table S15, namely Farnesol Oxidase XP_001606362.2 *D. ananassae*, this gene is not Farnesol Oxidase from *D. ananassae*, as stated in table S15. Rather, XP_001606362.2 is a PREDICTED: retinol dehydrogenase 12-like from *Nasonia vitripennis*. Again, I searched the

S. japonicum predicted proteins by BLASTP with XP_001606362.2 and I found no matches at all.

I believe that Suppl. Table 15 should be eliminated. Accordingly, the text on lines 317-318 as well as on lines 413-414 should be corrected to state that none of the enzymes involving the process of transferring farnesal to JH have been detected.

The authors insist in going swiftly through their proposal of an alternative JH synthetic pathway in *S. japonicum*, without making it clear that at this point there is absolutely no support for their hypothesis, from the RNA-seq or from the genome sequence prediction point of view, as far as I can see.

2) Throughout the entries in Tables S1, S2, S6, S8, S11, S13, S17 the authors refer to the Accession numbers of *Schistosoma japonicum* genes as Sjc_0021210, Sjc_0067200, etc. with the Spc_ prefix. I have not found in any database these accession numbers. Rather, Sjp_0021210, Sjp_0067200, etc. are found. I suggest that the authors use the appropriate Sjp_ accession numbers in all of the above supplementary tables.

Reviewers' comments:

Reviewer #2 (Remarks to the Author):

This is a re-submission of a paper by Wang et al describing gene expression patterns associated with reproduction in *Schistosoma japonicum*. The authors have addressed my previous concerns and made appropriate revisions. The revised manuscript will make an important, novel contribution to the field.

Response:

We thank the reviewer for the positive comments.

Reviewer #3 (Remarks to the Author):

In the revised version of the manuscript the authors have adequately addressed most of the questions and problems raised by the referees. Nevertheless, I still have two points that I believe need further attention.

Response:

Thank you for these comments and suggested changes, with which we concur will further improve the report.

1) I believe that the text describing the two possible enzymes in the JH pathway that the authors now mention in Supplementary Table S15 and in lines 317-318 still does not give a correct description of their findings.

First, the sentence on lines 315-318 start saying that “The RNAseq transcripts revealed a pathway of de novo synthesis of the JH precursor [...]” and continues to mention the finding of two possible enzymes in the last four steps of the pathway, citing Suppl. Table S15 with two genes that are annotated in the *S. japonicum* genome (Sjp_0058160.1..pep and Sjp_0036830.1..pep) without making it clear that there were no RNA-seq transcript reads from the present work matching these two genes predicted by the v4.0 genome (I could not find any denovo assembly entry in Suppl. Table S2 that was annotated as being similar to Sjp_0058160 or to Sjp_0036830).

Response:

From the de novo assembly of the transcriptome dataset, the candidate genes involved in the pathway of de novo synthesis of the JH precursor were identified. However, expression of genes involved in the conversion from farnesol to JH was not detected in RNAseq data. When the search was extended to genome data, two possible genes were found for enzymes in the last four steps of the pathway. So, for clarity, we have revised this section, as follows:

Line 311-315: “The RNA-seq transcripts supported a pathway of de novo synthesis of the JH precursor (farnesol) from acetyl-CoA (Supplementary Fig. 9). Furthermore, genes for two of four enzymes that participate in the conversion of farnesol to JH were detected from the genome sequences of *S. japonicum* (Supplementary Table 15). ”

Second and most important, the enzyme mentioned in Suppl. Table S15 (XP_011492954 *A. aegypti*) as being the Farnesal dehydrogenase is not the correct one.

The two Farnesal dehydrogenase genes from *A. aegypti* have been identified in 2013 by Rivera-Perez et al. (doi: 10.1016/j.ibmb.2013.04.002) as being Acc numbers AGI96742 and AGI96740, and I searched by BLASTP the protein database of *S. japonicum* for the presence of genes similar to those two Farnesal dehydrogenase genes with absolutely no success. There seems to be no gene similar to Farnesal dehydrogenase in the annotated *S. japonicum* genome. Also, there seems to be no denovo assembled transcripts for Farnesal dehydrogenase among the RNA-seq data of the authors, since I could not find any denovo transcript in Suppl. Table S2 that has been annotated as similar to AGI96742 and AGI96740 or to AAEL012161, AAEL012162 and AAEL012165, the original *A. aegypti* entries for these enzymes in the insect database.

Response:

In insects, the metabolism proceeds through four steps for the conversion of farnesol to juvenile hormone III (JHIII). 1) farnesol to farnesal by farnesol dehydrogenase (alternative name, farnesol oxidase, used in Zhu et al 2016^[1]); 2) farnesal to farnesoic acid by aldehyde dehydrogenase (alternative name, farnesal dehydrogenase, used in Zhu et al 2016^[1]); 3) farnesoic acid to methyl farnesoate by hormone methyltransferase; and 4) methyl farnesoate to JHIII by JH epoxidase. Here we performed BlastP with the mosquito farnesal dehydrogenases (AGI96742 and AGI96740)^[2] sequences as queries against the Schistosoma database in NCBI and readily detected matches to highly conserved sequences of schistosomes (Fig 1 and 2 below), which indicated the presence of functional farnesal dehydrogenase(s) in schistosomes.

1. Zhu, J. *et al.* Mevalonate-farnesal biosynthesis in ticks: Comparative synganglion transcriptomics and a new perspective. *PLoS ONE* 11, e0141084 (2016).
2. Rivera-Perez, C. *et al.* Aldehyde dehydrogenase 3 converts farnesal into farnesoic acid in the corpora allata of mosquitoes. *Insect Biochem Mol Biol* 43, 675-82 (2013).

Fig 1. BlastP result of farnesal dehydrogenases (AGI96742) against the *Schistosoma* database in NCBI

AGI96740:aldehyde dehydrogenase 3-1 variant..

RID 2KY6U544014 (Expires on 11-16 00:48 am)

Query ID AGI96740.1
Description aldehyde dehydrogenase 3-1 variant C [*Aedes aegypti*]
Molecule type amino acid
Query Length 495

Database Name nr
Description All non-redundant GenBank CDS translations+PDB+SwissProt+PIR+PRF excluding environmental samples from WGS projects
Program BLASTP 2.5.1+ Citation

Other reports: Search Summary Taxonomy reports Distance tree of results Multiple alignment

New Analyze your query with SmartBLAST

Graphic Summary

+ Show Conserved Domains

Distribution of 29 Blast Hits on the Query Sequence

Mouse over to see the title, click to show alignments

Descriptions

Sequences producing significant alignments:

Select: All None Selected:0

Alignments Download GenPept Graphics Distance tree of results Multiple alignment

	Description	Max score	Total score	Query cover	E value	Ident	Accession
[ ]	putative Fatty aldehyde dehydrogenase [Schistosoma japonicum]	387	387	92%	9e-131	41%	CAX73865.1
[ ]	putative Fatty aldehyde dehydrogenase [Schistosoma japonicum]	387	387	92%	2e-130	41%	CAX73864.1
[ ]	putative aldehyde dehydrogenase [Schistosoma mansoni]	343	343	74%	6e-115	46%	XP_018654064.1
[ ]	Fatty aldehyde dehydrogenase [Schistosoma haematobium]	267	267	65%	1e-85	40%	XP_012797550.1

Fig 2. BlastP result of farnesal dehydrogenases (AGI96740) against the *Schistosoma* database in NCBI

Also, regarding the entry for the supposed second insect gene in the JH pathway, given in Table S15, namely Farnesol Oxidase XP_001606362.2 *D. ananassae*, this gene is not Farnesol Oxidase from *D. ananassae*, as stated in table S15. Rather, XP_001606362.2 is a PREDICTED: retinol dehydrogenase 12-like from *Nasonia vitripennis*. Again, I searched the *S. japonicum* predicted proteins by BLASTP with XP_001606362.2 and I found no matches at all.

Response:

The XP_001606362.2 sequence is indeed from *Nasonia vitripennis* rather than *D. ananassae*. However, it is a predicted farnesol oxidase (farnesol dehydrogenase) as noted in Zhu et al 2016^[1]. We also run BlastP using a functionally confirmed NADP⁺-dependent farnesol dehydrogenase^[2] (D2WKD9) from *A. aegypti* and found matches in schistosomes (see Fig 3 below).

1. Zhu, J. et al. Mevalonate-farnesal biosynthesis in ticks: Comparative synganglion transcriptomics and a new perspective. *PLoS ONE* 11, e0141084 (2016).
2. Mayoral, J.G., Nouzova, M., Navare, A. & Noriega, F.G. NADP⁺-dependent farnesol dehydrogenase, a corpora allata enzyme involved in juvenile hormone synthesis. *Proc Natl Acad Sci U S A* 106, 21091-6 (2009).

sp|D2WKD9|SDR1_AEDAE Farnesol dehydrogenase...

RID 2M8BR06M014 (Expires on 11-16 03:41 am)

Query ID |d|Query_346037
 Description sp|D2WKD9|SDR1_AEDAE Farnesol dehydrogenase OS=Aedes aegypti GN=SDR-1
 Molecule type amino acid
 Query Length 245

Database Name nr
 Description All non-redundant GenBank CDS translations+PDB+SwissProt+PIR+PRF excluding environmental samples from WGS projects
 Program BLASTP 2.5.1+ Citation

Other reports: Search Summary Taxonomy reports Distance tree of results Multiple alignment

New Analyze your query with SmartBLAST

Graphic Summary

Show Conserved Domains

Distribution of 28 Blast Hits on the Query Sequence

Color key for alignment scores

- <40
- 40-50
- 50-80
- 80-200
- >=200

Descriptions

Sequences producing significant alignments:

Select: All None Selected:0

Alignments Download GenPept Graphics Distance tree of results Multiple alignment

Description	Max score	Total score	Query cover	E value	Ident	Accession
[ ] SJCHGC06279 protein [Schistosoma japonicum]	78.2	78.2	81%	4e-17	29%	AAW26955.1
[ ] putative tropinone reductase [Schistosoma mansoni]	74.3	74.3	81%	1e-15	29%	XP_018651437.1

Fig 3. BlastP result of D2WKD9 against the *Schistosoma* database in NCBI

I believe that Suppl. Table 15 should be eliminated. Accordingly, the text on lines 317-318 as well as on lines 413-414 should be corrected to state that none of the enzymes involving the process of transferring farnesal to JH have been detected.

The authors insist in going swiftly through their proposal of an alternative JH synthetic pathway in *S. japonicum*, without making it clear that at this point there is absolutely no support for their hypothesis, from the RNA-seq or from the genome sequence prediction point of view, as far as I can see.

Response:

As mentioned above, our RNA-Seq data revealed a pathway of de novo synthesis of the JH precursor (farnesol) and the public schistosome genome data indicated two possible genes involved in the first two steps for the conversion of farnesol to JH, suggesting that there is a potential for JH biosynthesis. This information in combination with the previous reports of ecdysone, ecdysone-induced protein 78 and allatostatin in schistosomes, as well as allatostatin receptor like protein in our RNAseq dataset, led us to propose that blood fluke likely uses a system similar to that in insects to regulate its development. Hence, we have modified the contents in Suppl. Table 15 and rewritten the text (see lines 311-315; lines 408-411) as follows:

Line 311-315: “The RNA-seq transcripts supported a pathway of de novo synthesis of the JH precursor (farnesol) from acetyl-CoA (Supplementary Fig. 9). Furthermore, genes for two of four enzymes that participate in the conversion of farnesol to JH were detected in the genome sequences of *S. japonicum* (Supplementary Table 15) ”

Line 408-411: “We detected a de novo synthesis pathway of JH precursor (farnesol) from acetyl-CoA (Supplementary Fig. 9) from our RNA-seq data, and further detected two possible genes involved in the first two steps for the conversion of farnesol to JH from the genome data of S. japonicum (Supplementary Table 15);”

2) Throughout the entries in Tables S1, S2, S6, S8, S11, S13, S17 the authors refer to the Accession numbers of Schistosoma japonicum genes as Sjc_0021210, Sjc_0067200, etc. with the Sjc_ prefix. I have not found in any database these accession numbers. Rather, Sjp_0021210, Sjp_0067200, etc. are found. I suggest that the authors use the appropriate Sjp_ accession numbers in all of the above supplementary tables.

Response:

Thank you for this comment. We have replaced the “Sjc” prefix for accession numbers with “Sjp”.

REVIEWERS' COMMENTS:

Reviewer #3 (Remarks to the Author):

The authors have adequately responded to my queries and have changed the main text and Supplementary Table S15.